# Wearable and Flexible Sensor Devices: Recent Advances in Designs, Fabrication Methods, and Applications

**DOI:** 10.3390/s25051377

**Published:** 2025-02-24

**Authors:** Shahid Muhammad Ali, Sima Noghanian, Zia Ullah Khan, Saeed Alzahrani, Saad Alharbi, Mohammad Alhartomi, Ruwaybih Alsulami

**Affiliations:** 1Department of Engineering and Technology, School of Computing and Engineering, University of Huddersfield, Queensgate, Huddersfield HD1 3DH, UK; 2Engineering Department, The City of Liverpool College, Liverpool L3 6BN, UK; 3CommScope Ruckus Wireless, 350 W Java Dr, Sunnyvale, CA 94089, USA; sima_noghanian@ieee.org; 4National Physical Laboratory, Electromagnetic & Electrochemical Technologies Department, Teddington TW11 0LW, UK; zia.khan@npl.co.uk; 5Department of Electrical Engineering, University of Tabuk, Tabuk 71491, Saudi Arabia; saeedalzahrani@ut.edu.sa (S.A.); malhartomi@ut.edu.sa (M.A.); 6King Abdulaziz City for Science and Technology, Riyadh 11442, Saudi Arabia; saalharbi@kacst.gov.sa; 7Department of Electrical Engineering, Umm Al-Qura University Makkah, Mecca 24382, Saudi Arabia; rrsulami@uqu.edu.sa

**Keywords:** wearable devices, DIW and DW fabrication methods, WBAN

## Abstract

The development of wearable sensor devices brings significant benefits to patients by offering real-time healthcare via wireless body area networks (WBANs). These wearable devices have gained significant traction due to advantageous features, including their lightweight nature, comfortable feel, stretchability, flexibility, low power consumption, and cost-effectiveness. Wearable devices play a pivotal role in healthcare, defence, sports, health monitoring, disease detection, and subject tracking. However, the irregular nature of the human body poses a significant challenge in the design of such wearable systems. This manuscript provides a comprehensive review of recent advancements in wearable and flexible smart sensor devices that can support the next generation of such sensor devices. Further, the development of direct ink writing (DIW) and direct writing (DW) methods has revolutionised new high-resolution integrated smart structures, enabling the design of next-generation soft, flexible, and stretchable wearable sensor devices. Recognising the importance of keeping academia and industry informed about cutting-edge technology and time-efficient fabrication tools, this manuscript also provides a thorough overview of the latest progress in various fabrication methods for wearable sensor devices utilised in WBAN and their evaluation using body phantoms. An overview of emerging challenges and future research directions is also discussed in the conclusion.

## 1. Introduction

Wireless body area networks (WBANs) are used in monitoring physical activities, navigation and searching, healthcare, and defence, as shown in Figure 1. WBAN relies on wearable devices to provide a possible solution to clinical diagnostics using physical, chemical, and biological sensors to monitor vital body signals in real-time. These devices can be integrated into wearable materials such as glasses, jewellery, face masks, wristbands, watches, fitness bands, tattoos, bandages, patches, and wearable logos that can be used in both non-invasive and invasive applications [1]. Flexible and stretchable devices are available that can adapt to any surface without breaking and have the capability to elongate or deform under stress without losing functionality. They are usually made from an elastic; flexible substrate that has active sensing components bonded to it and can be employed in numerous applications. Recently, a self-powered and repairable humidity sensor constructed from silk nanofibre (SNF) film advanced the development of wearable technology by offering novel approaches to the domains of intelligent control, health monitoring, and next-generation wearable technology, including self-powered triboelectric nanogenerator (TENG) sensors utilising polyethylene oxide and copper-oxide nanofibres to monitor humidity as well as ethanol in simulated exhalation [2,3]. A range of wearable devices has been developed incorporating different sensing applications. These devices utilise systems-on-chips (SoCs) to combine the sensing and processing of bio signals [4]. The most current developments in skin-worn, tattoo-based wearable electrochemical devices, electrolyte and metabolite sensors, biofuel cells, batteries, and nanomaterial-based wearable skin devices were investigated [5]. The application of liquid metal or its alloys as printing inks holds great potential in the realm of flexible electronics. It presents a convenient and direct approach to fabricating electronic components by simply writing them out using these liquid materials. This innovative method involves incubating the liquids to create the desired electronic structures [6]. Within the context of the DIW method, there is a comprehensive evaluation of crucial components for wearable electronic devices. These components include nanogenerators for energy generation, lithium–ion batteries for energy storage, and strain sensors for managing energy consumption. The focus of this examination revolves around assessing their performance metrics and exploring various fabrication techniques [7]. Shoe-based sensor systems can handle various vertical ground reaction forces (vGRFs), motion intent learning, plantar pressure, and fall detection, and they are useful in the calculation of the centre of mass displacement while walking, as well as pedestrian navigation and tracking [8]. A loop-based sensing antenna was designed using NinjaFlex material on a 3D hemispherical-shaped substrate [9]. The printing method was fused deposition modelling (FDM). An ultra-wideband (UWB) monopole-sensing antenna was designed on a flexible substrate and printed using the fuse-based filament method [10]. Another paper reports a new radio frequency identification (RFID) tag was printed on a cotton fabric using graphene ink via an nScrypt 3D-DW dispensing system [11]. The recent progress in flexible electronic systems, which utilises transistor-based circuitry and active-matrix technology built on plastics or textile substrates, has been extensively reviewed [12]. The focus lies on fabrication techniques using conventional materials that are commonly used in electronics. These flexible systems have paved the way for the development of various wearable sensor devices, including tattoo-based and biofluidic prototypes. These sensors have found applications in healthcare, particularly in WBAN systems [13].

However, there has been a significant surge in the development of sensing systems for industrial electronics. These sensing devices have undergone design, prototyping, and testing processes in both laboratory and real-life scenarios [14]. To cater to different electrical, mechanical, and thermal requirements, a wide range of flexible sensor devices has been developed using materials such as rigid, flexible, liquid, and single-crystal silicon, as well as a variety of polymers and nanomaterials, such as polydimethylsiloxane (PDMS), polyethylene terephthalate (PET), polyimide (PI), and polyethylene–naphthalate (PEN). Several flexible sensor devices have been designed using conductive polymer types of nanocomposite materials, such as poly (3,4-ethylene dioxythiophene), polystyrene sulfonate (PEDOT: PSS), and polyaniline. In addition, carbon-based allotropes have been employed as processed materials to design wearable and flexible sensor devices [15]. A new stretching-insensitive bending sensor was designed and tested in [16]. A thin, porous PDMS sponge with a CNT network coating (CCPPS), which is flexible and bending-insensitive, was integrated with two electrodes to monitor the human body [17]. To support or use soft active materials such as carbon, metal, ion, or liquid metals, researchers have reported using flexible/stretchable non-active materials, which are discussed in [18], such as a small contact-lens platform that can be used for various human–machine interactions (HMI) [19]. Several human movements are sensed by shoe-mounted wearable sensors, which may be used for various purposes, including activity detection and energy expenditure estimation [20]. A disposable, ultrathin multi-layer adhesive patch with electrodes and interconnects for skin interfacing based on an Ag-In-Ga-filled elastomer was digitally produced and reported [21]. Nanomaterials have made it possible to develop tactile sensors that are well-suited for applications involving body parts such as skin. However, these sensors often encounter issues with a noisy environment and low power signals, which may contain contaminated data. The presence of noise can hinder the accurate collection of relevant information. Considering this, researchers have suggested the adoption of signal-processing methods based on neural networks (NNs) to improve the signal-to-noise ratio (SNR) and improve signal accuracy. Such approaches can help mitigate the impact of noise and enhance the reliability and accuracy of the collected data [22]. An advancement in machine learning combined with wearable technologies has revolutionised medical monitoring and human–machine interaction (HMI) by leveraging the following four key bio signals: electrocardiograms (ECGs), electromyography (EMG), electroencephalography (EEG), and photoplethysmography (PPG). ECG, when integrated with machine learning, enables accurate disease detection, heartbeat classification, and real-time cardiac monitoring, enhancing cardiac healthcare and identifying conditions like sleep apnea and stroke risk. EMG-driven machine learning has transformed human–machine interaction (HMI) applications, such as virtual reality interfaces and prosthetics, while also aiding in the diagnosis of neuromuscular diseases. EEG, powered by machine learning, facilitates brain–computer interface development and cognitive state analysis, benefiting psychology and neurology. Meanwhile, PPG, with its non-invasive nature and enhanced signal processing, supports the diagnosis of cardiovascular conditions and real-time health monitoring [19,22]. With the development of biodegradable materials, there is now a great chance to revolutionise healthcare technologies using sensors that will naturally break down after use. With the recent development of superhydrophobic wearable sensors, it is now evident that wearable devices with superhydrophobic functions have significant potential for diverse applications [23]. MXene ink (transition metal carbides, nitrides, and carbonitrides) has penetrated such diverse fields of research as wonder materials. The most attractive features of their applications show great promise in energy storage conversion, harvesting, and sensing [24].

E-skins with multimodal sensing capabilities have been extensively reviewed, highlighting their potential for diverse applications, including health monitoring, intelligent prosthetics, HMIs, and robotic systems. Further, these multimodal e-skin systems are classified into three modes, as follows: (1) integration of multiple physical sensors, (2) integration of physical and electrophysiological sensors, and (3) integration of physical and chemical sensors. These systems primarily mimic human skin functions by detecting physical stimuli like force, strain, vibration, temperature, and humidity while also integrating additional sensing modalities, such as ultra-violet (UV) light sensors, to create next-generation smart skins [5]. These advancements have shed light on the future direction of next-generation wearable sensor devices. This evolving trend encompasses various areas, including the applications of sixth generation (6G) radio frequency (RF) in sensing, imaging, and healthcare. These applications encompass a broad spectrum, ranging from wireless networks employed in neural implants and drug delivery systems to guaranteeing safety concerns related to exposure to microwaves. The exploration of these avenues paves the way for innovative developments in wearable technology [25]. In wearable applications, specific absorption rate (SAR) measures the rate at which the human body absorbs energy from electromagnetic fields (EMF), with regulatory limits set to prevent harmful exposure. In the U.S., the Federal Communications Commission (FCC) has set the SAR limit at 1.6 W/kg, averaging over 1 g of tissue for the head, and 4.0 W/kg, averaging over 10 g for extremities. The International Commission on Non-Ionizing Radiation Protection (ICNIRP) has slightly different guidelines, with 2.0 W/kg for head exposure in the EU, Japan, and China. In overall wearable and flexible sensor devices, accurate SAR measurement and adherence to these regulatory limits are essential for ensuring user safety, particularly with newer high-frequency applications. Updating safety guidelines to account for prolonged exposure and new frequency ranges is increasingly important as these devices become more integrated into daily life. To minimise the effects of microwave wave radiation, various methods such as reducing direct exposure, maintaining distance from radiation sources, using protective layers like shielding, and period structures can be used [25,26].

In WBAN, additive manufacturing (AM) methods have undergone significant advancements over the past few decades, catering to diverse applications such as medical devices and orthopaedic equipment [27]. These advancements have opened new possibilities for the manufacturing and utilisation of wearable sensor devices within the WBAN domain using various inks or liquids. For example, DW is an additive manufacturing (AM) method that directly transfers onto a substrate material, also called digital writing or digital printing, while DIW, called the direct-write assembly, is an extrusion-based 3D AM printing method in which liquid or semisolid colloidal inks are dispensed via small nozzles under controlled pressure and flow rates. These methods are fabrication methods that are used to produce micro devices, printed circuit board (PCB) electronics, optics biosensing devices including biocompatible scaffolds for tissue engineering, and high-performance energy storage systems [27,28]. In [29], the research investigated a liquid-based eutectic gallium–indium (EGaIn) to design sensing devices in WBAN. Intelligent sensing based on commercial Wi-Fi devices is an emerging technology for next-generation healthcare monitoring with great potential for future development due to its low cost and availability. However, an integration of TENG technology presents a promising solution for wearable health devices and disease diagnostics, as it allows for real-time, portable monitoring without the need for external power sources. Key characteristics of the sensors created in this manner include their biocompatibility, scalability, lightweight design, flexibility, and wearability. Additionally, connection with a triboelectric source has made it possible to develop a self-powered, wearable, portable sensor. Smart sensors are useful because they can be created rapidly and affordably, worn conveniently, have a long lifespan, react swiftly, and may be used anywhere and at any time. They are also resistant to water and other factors [30,31]. However, this manuscript aims to address these gaps by considering the following key requirements:Exploration of the latest developments in flexible, rigid, liquid, and next-generation devices.Examination of advanced DIW and DW fabrication methods.Integration of body phantom models to account for the impacts of the human body and development of more accurate and realistic human body phantom models.

By incorporating these requirements, this review paper offers a unique view that simultaneously considers three dynamic developments that have not been extensively studied together. Many researchers are currently involved in solving all these challenges. For example, several different types of wearable and flexible sensor devices have already been discussed and have shown a very reasonable performance in WBAN systems such as flexible and stretchable, self-powered and degradable humidity sensors, biofluidic, Mxene ink, liquid, and next-generation miniaturised wearable and flexible sensor devices [8,13,15,17]. State-of-the-art reviews based on advancements in DW- and DIW-based and emerging fabricated systems such as 3D to 6D have also been studied [27,28]. The human body methods such as on-body body phantom models for evaluation have already been drawn with the aim to provide researchers with a holistic understanding of devices and their impact on complex human body situations and provide advancements in design and manufacturing methods. Therefore, the development of realistic simulation and emulation platforms is essential for evaluating these techniques in their initial phase of development. In recent years, the research community has shown a keen interest in the development of more accurate and realistic phantoms for different sensing techniques based on microwaves, magnetic resonance imaging (MRI), optics, and acoustic optics [11,12,13,14,30]. Given this, it is fruitful to retrospectively examine the recent progress in the field of wearable and flexible sensor devices, including their design, fabrication, and on-body applications in WBAN systems. This manuscript is arranged as follows: It begins with an introduction, followed by Section 2, which describes wearable and wireless communication devices. Section 3 provides the latest comprehensive overview of DIW and DW fabrication techniques for wearable devices, featuring examples and detailed descriptions. Section 4 focuses on the impact on the human body to evaluate the performance of wearable devices through simulation and measurement. Lastly, Section 5 concludes the review and offers recommendations for future directions.

## 2. Wireless Communication Technologies

Wireless communication systems offer a versatile solution for various communication needs, providing a viable alternative to expensive and inflexible cable network systems. These wireless communication systems can be categorised into different types based on their range and transmission speed, including short-range, cellular, and long-range networks. Furthermore, they include outdoor and indoor positioning systems, further highlighting their diverse applications and capabilities [32,33]. Indoor environments pose a significant challenge for navigation systems, as they account for up to 80% of human activities. While reliable and widely used outdoor navigation systems like “Global Positioning Systems (GPS), the Global Navigation Satellite System (GLONASS), and the Bei Dou Navigation Satellite System (BDS)” offer precise location services, their accuracy diminishes indoors due to signal losses caused by buildings, the multipath effect, and time-delay issues. Consequently, the demand for positioning technologies falls short of meeting the requirements of an indoor system. To address this, an Indoor Positioning System (IPS) was developed which continuously and in real-time provides accurate locations of people or objects within enclosed spaces. IPSs serve multiple purposes, including object detection and tracking, assisting the daily routines of the elderly and disabled, and enabling medical and vital signs monitoring during emergencies. Indoor positioning services are crucial in various public areas, as they assist visitors in finding their desired destinations and offer navigation support for individuals who are blind or have impaired mobility. Moreover, hospitals may use IPSs for effective patient tracking [34]. Indoor positioning systems, including hardware platforms and localisation algorithms, have received great attention. Several sensing technologies, such as RFID, Wi-Fi, acoustic signals, and Bluetooth, have emerged as viable options. Different parameters can be used to create various subcategories within these IPSs. One common categorisation is based on the types of sensors employed. These include camera-based, infrared-based, tactile, polar, sound-based, Wi-Fi and WLAN, RFID systems, UWB, Assistant GNS, pseudo-based satellite systems, as well as other RF-based devices such as ZigBee, Bluetooth, digital television, cellular-based networks, radar, inertial-based navigation, and magnetic-based systems. This diverse range of sensor technologies contributes to the wide array of options available for indoor positioning and tracking purposes. Table 1 illustrates the available wireless technologies that are used in the literature for several applications.

In [35,36], researchers evaluated power consumption and the quality of the data exchange in WBAN systems. RFID plays a significant role in numerous applications within short-range wireless networking. Although not as commonly used as RFID, Near Field Communication (NFC) is compatible with smartphones and tablets. In addition to Wi-Fi for short-range high-speed communication, cellular wireless systems like GSM (Global System for Mobile Communications), GPRS (General Packet Radio Service), and 3G, 4G, and 5G (third-, fourth-, and fifth generation) wireless technology for mobile telecommunication enable long-range wireless connectivity. GPRS-based systems are employed in various cellular wireless communication networks. Various researchers employed a design based on Wi-Fi and GPRS to monitor multiple wireless communication applications, achieving minimal data loss. For long-range networks, LPWANs (low-power wide-area networks) have been specifically designed to be compatible with the Internet of Things (IoT) devices. LPWANs offer low data rates and energy consumption, starting from 2G and extending to 5G networks [46,47,48]. The principle of free hardware has recently been implemented in various wireless applications to control and analyse data. However, these technologies have not yet been fully utilised in healthcare, even though they are gaining traction due to their compatibility with Bluetooth, Wi-Fi, ethernet, LoRa (long-range), and Lora WANs (long-range wide-area networks). One example is ESP8266EX, which boasts a combination of affordability, energy efficiency, compactness, and dependable performance [54]. 

### Smart Wearable Devices

The global phenomenon of population aging, coupled with concerns surrounding family planning and its impact on birth rates, has led to significant implications for the socio-economic healthcare system. The cost of medicine, medical devices, and hospital care continues to rise. To address these challenges, wearable biosensors have emerged as non-invasive smart gadgets that enable continuous real-time patient monitoring. These technologies serve as effective tools for disease management, promoting healthier behaviours, and facilitating early identification of risk factors, diagnosis, and treatment [55,56]. Sensing electronic devices attached to soft tissue and worn as skin patches have recently been developed to establish a new interface for monitoring applications, as shown in Figure 2a [57]. Skin patches, when placed discreetly beneath clothing, can capture highly precise data without being influenced by human movement. These patches are capable of accurately monitoring vital signals like temperature, tension, sweat, and heart rate when affixed to the surface of the skin [58]. Sweat holds significant importance as a crucial bodily fluid due to its composition of essential elements such as electrolytes, small chemicals, and proteins. In recent years, the development of sweat-analysis devices has enabled the detection of various components present in sweat. These devices utilise sensing techniques to conduct a comprehensive analysis of sweat composition [59]. 

It is crucial to monitor changes in skin temperature during the initial phases of diagnosis and treatment [60]. Recently, a group of researchers developed a flexible stress-monitoring patch with a small area of skin contact. This is an important factor since it reduces the effects of the patch on the human body. In [61], it is reported that a group of scientists created a photonic crystal for non-invasive glucose monitoring of tear fluid made of colloidal particles dissolved in the hydrogel in a face-centre cubic structure. Tear fluid had 100 mol/L or more of glucose, which was detected by photonic-based glucose-sensing materials. The detection threshold for tear fluid was 1 mol/L. A mouthguard, which uses built-in wireless electronics to test salivary uric acid, is a recent addition to wearable devices in WBAN systems [62]. In another research investigation, a mouthguard was used to monitor and detect salivary glucose [63]. To measure blood pressure (BP) without a cuff, a portable patch sensor with an adjustable flexible piezo-resistive sensor (FPS), as well as epidermal ECG sensors, has been created [64]. Numerous studies have investigated wearable flexible and fluidic devices used to measure vital signals [65]. Wearable devices have found applications beyond human health and well-being, extending to animal health in the field of pet care and animal husbandry, etc [66]. Autonomously functioning wearable sensor systems have the capability to integrate point-of-care functionality with mobile connectivity. These systems comprise a combination of active and passive components, as well as sensing devices. These autonomously functioning wearable sensor systems support ongoing evaluation for disease prevention, early reduction of health risk factors, and maintenance of optimal, lifetime health quality [66,67]. However, modern wearable devices perform high-quality measurements as compared to conventional medical devices. The first-generation wearables predominantly focused on biophysical devices designed to monitor and track metrics related to physical activity, such as heart rate and body temperature. These devices were integrated with various wearables such as watches, shoes, and headphones [68]. The initial wave of wearables gained widespread adoption, leading to a shift in focus toward second-generation devices. The second-generation wearables encompass non-invasive or minimally invasive biochemical and multi-functional designs. Second-generation devices encompass a range of options, including invasive microneedles, injectable devices, and compact alternatives such as skin patches, tattoos, tooth-mounted sensors, and contact lenses. These advancements offer diverse solutions for monitoring and enhancing various aspects of human health and well-being [69]. A group of second-generation devices utilise biofluids and employ biorecognition components to convert analytes into measurable signals. These devices are typically laboratory-based and include examples such as the Freestyle Libre for glucose monitoring and sweat-based patches for Gx detection. Numerous wearable biochemical and biophysical sensor devices have been utilised for wellness applications, including the detection of COVID-19 [70]. In [71], a novel nanostructured smart tactile sensing device was constructed using 3D printing with an integrated electronic component. A tactile sensing device was improved using neural networks (NN) on a digital unit. Various telehealth devices are employed to identify an illness of an individual. By utilising four essential bio signals such as ECG, EMG, EEG, and PPG, machine learning and wearable technology have transformed medical monitoring and HMI. When machine learning is combined with ECG, it can be used to accurately detect diseases, classify heartbeats, and provide real-time cardiac monitoring. This improves cardiac care and helps identify disorders like stroke risk and sleep apnea. EMG-driven machine learning has revolutionised HMI applications and helped diagnose neuromuscular illnesses. Examples of these applications include virtual-reality interfaces and prosthetics. Machine learning-powered EEG helps with cognitive state analysis and brain–computer interface development, which is beneficial in neurology and psychology. PPG, on the other hand, facilitates real-time health monitoring and the identification of cardiovascular disorders due to its non-invasive nature and improved data processing. Therefore, machine learning has revolutionised wearable and HMI applications and helped in diagnosis [19,22,72]. 

Researchers integrated a contact lens device based on a multifunctional ultrathin MoS2 transistor, as shown in Figure 2b [73]. In [74], a hard-structure sensor, along with circuit chips, was incorporated into the lens-type substrate and contact-tear fluid method via a microfluidic-based sensing channel. An ultrathin-type sensor was fixed onto the lenses and came into contact with the tear fluid, which provided detection without interfering with blinking. Soft electronic devices offer an excellent interface with the human body’s skin, enabling real-time monitoring of vital signals [75]. Researchers investigated smart glasses and head-mounted computers capable of displaying information [76]. They employed nanomaterials with mechanical structures and stretchable polymers to develop electronic devices that are both flexible and stretchable, enabling seamless integration with living organisms. These devices are capable of detecting parameters such as temperature, ion concentrations, and electrocardiogram (ECG or EKG) signals, in vivo or in vitro [77]. Several transdermal patches were used for pharmaceutical and chemical agents through the human body’s skin [78]. A single-walled carbon nanotube (SWCNT) was designed to monitor glucose oxidase–Nafion composites for sensing [79]. Recently, hybrid devices based on metal, dielectric, and metallic polymers were proposed, and their performance was simulated for various frequency bands to design wearable and flexible devices. A group of researchers recently investigated a self-powered device that can easily harvest ambient energy to run a sensor, removing the need for external power [80]. Piezo-electric nano-generators (PENGs) and tribo-electric nano-generators (TENGs) have predominantly been employed in the development of self-powered sensor devices. These wearable sensor devices play a crucial role in measuring vital signals such as respiratory rate, temperature, and blood-oxygen levels, which are essential in accurately diagnosing and monitoring COVID-19 [81]. High-performance wearable electrical devices require fibrous material with high strength and considerable stretchability to provide durability and stability. An ultra-thin, robust, and extensible conducting microfibre is constructed for designing fibrous-type mechanical sensors for wearable applications. These sensing devices demonstrated high sensitivity in detecting strains, with a high resolution and a large detection range simultaneously [82].

**Figure 2 sensors-25-01377-f002:**
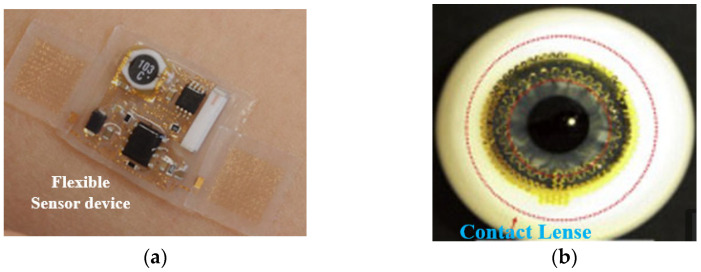
Wearable electronic devices (**a**) soft tissue [57], and (**b**) contact lens [74].

In [83], researchers evaluated traditional polysomnography measurement tools to study sleep. Some of these tools lack comfort and impose high additional costs. As an alternative, several innovative designs, such as jewellery and body-mounted sensing devices, have been developed as non-invasive options to measure vital signals during sleep. They provide a more comfortable and cost-effective solution for monitoring sleep-related activities and signals [84]. Sleep disorders may be controlled through advanced technologies that make early diagnosis possible [85]. MXenes ink is a highly conductive material [86], as highlighted, and applications have been found in various flexible and stretchable devices. In [87], a highly integrated multipurpose sensor device for intracorporeal applications was proposed. This intracorporeal system incorporated temperature, capacitive-type pressure, and electrochemical-based oxygen sensors onto a single chip, along with their corresponding interface application-specific integrated circuits (ASICs). A low-cost UWB-based landslide monitoring device, named Wi-GIM, was introduced in [88]. This device utilises UWB technology to determine the relative distances between monitoring points and effectively track ground deformations associated with landslides. A novel conformable sensor interface was created in [89]. This sensor was designed to be easily attached and detached from face masks. Smartwatches have become a well-established tool for continuous activity and fitness tracking. Longitudinal tracking using smartwatches is being utilised in clinical applications, including remote monitoring of Parkinson’s disease (PD), which is the most prevalent neurodegenerative disease [90]. In [91], researchers investigated biocompatible and biodegradable materials for flexible and stretchable devices without producing any environmental problems. As reported in [92], a laser-engraved wearable sensor was used for the sensitive detection of uric acid and tyrosine in sweat, as shown in Figure 3a. Extensive research has been conducted on intricate measurements, including ECG. For instance, the Apple watch screen has been investigated to detect atrial fibrillation simply through the touch of a finger [93]. Soft polymers have been explored as electrodes and active layers to produce flexible and stretchable electronic devices for wearable applications [94]. Various HMI-based (human–machine interface) vision assistance devices are now possible thanks to advancements in micro-electronics and nanofabrication for wearable and flexible materials. These applications include integrating sensors, circuits, and components into a small contact-lens design. Its interaction with apps for displaying information, detecting eye movements, recovering vision, and identifying specific biomarkers in tear fluid is made possible through various smart contact-lens materials, device designs, and componentry [95]. Google’s Fitbit-based tracker system examines blood-oxygen levels [96]. Recently, a four-layer sensor device was used for the healthcare system, which is compatible with 5G networks [97]. 

In another study [98], researchers proposed a sensor network that harnesses 5G-wireless networks for healthcare and safety applications. The combination of 5G-wireless technology and machine-to-machine (M2M) communications enables the continuous monitoring of chronic diseases. Thus, M2M systems are seamlessly integrated with 5G-enabled devices to provide ongoing monitoring for chronic diseases. The design of an e-health system tailored for chronic patients was explored by studying the infrastructure and protocols of 5G networks [99]. Continuous data transfer and sensing applications were facilitated using 5G-cellular channels as highlighted [100]. The integration of IoT, machine learning, data analytics, and 5G and beyond-5G (B5G) networks have opened new ways for innovative diagnostic methods that have the potential to lower hospital and medical costs. Furthermore, 5G technology brings several advantages to robotic surgery care by providing faster download rates, video calls, and gaming capabilities [101]. Advancements in flexible and stretchable technology have also enabled the utilisation of implantable systems in various areas, including the deep brain, intravascular space, intracardiac space, and even the interior structure of a single cell [102]. Several new sensing devices have been designed; for example, a degradable, tuneable graphene-based antenna and a band stop filter/antenna sensor have been reported [103]. In [104], a 3D curved-shaped antenna was reported, featuring a curved layout and a non-radiating edge gap-coupled configuration, suitable for diverse applications. Furthermore, a lightweight and scalable flexible integrated circuit phased array system has been designed which can be conveniently stored or rolled in a limited space, as shown in Figure 3b [105]. In recent developments, a multi-material approach has emerged as a viable solution to enhance the safety of wireless, skin-based interfaced bioelectronic sensors. This approach, depicted in Figure 3c, utilises a combination of different materials to ensure improved safety and robust functionality of wearable and flexible devices [106]. Utilising biodegradable, implantable, and wearable sensors for healthcare could change the future of disease management in healthcare toward preventative, predictive, and personalised management of diseases. Furthermore, the manufacturing of biodegradable electronics and sensors that naturally dissolve under ambient conditions could mitigate electronic waste challenges. The degradation rates of materials and dissolution behaviour of bioresorbable devices in different biological systems can vary significantly. Furthermore, developing a high-performance and miniaturised power supply, circuits, and chips on biodegradable substrates and wireless transmission is required to develop a fully biodegradable platform. Among several options, biodegradable sensors (BD-SENs) have proven to be the best option and are widely used in the food and medical industries. Moreover, BD-SENs are widely used in implanted medical devices, which disintegrate into the body through various degradation mechanisms once the intended function is completed. The use of these implantable BD-SENs helps to avoid the need for additional sensor removal operations. Implantable sensors with controlled degradation mechanisms will also aid in determining how long the device should operate inside the body. The issue of electronic waste has been addressed by the design of these environmentally friendly and sustainable sensors, which also serve as a green substitute for traditional non-biodegradable sensors. Degradable piezoelectric biomaterials have recently attracted more attention as potential medicinal applications. They provide a self-sufficient solution using biomechanical energy harvesting. This promising path toward the development of cutting-edge, transient biomedical devices that support both improved patient care and real-time monitoring is made possible by the combination of degradable sensors with flexible, piezoelectric materials. Future implantable sensors need to use as little power as possible, for example, getting power from the body’s movement, heat, or other in situ energy sources such as biofluid [91,103]. In [107], flexible MEMS-type sensor devices were proposed. For example, MEMS technologies have been used to produce such flexible electronics, they need equipment for vacuum deposition, photolithography, and wet and dry etching, as shown in Figure 3d.

**Figure 3 sensors-25-01377-f003:**
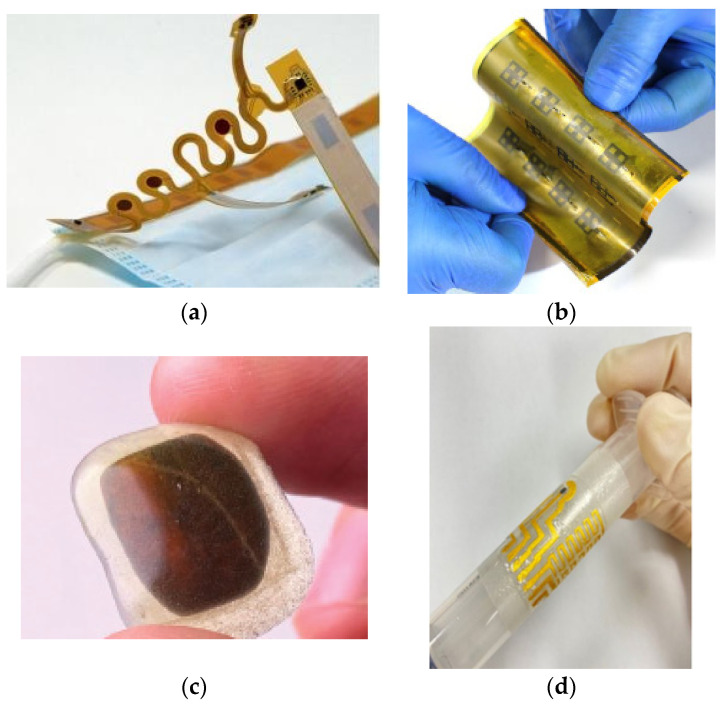
Smart devices: (**a**) laser-engraved [92], (**b**) flexible [105], (**c**) multi-materials devices [106], and (**d**) MEMS sensor [107].

Various advanced flexible electrochemical sensor devices have been used for various applications, such as in fitness and biomedicine [108]. With significant advancements in wearable devices, there has been an exploration of near-infrared spectroscopy (NIRS) electronics, which hold potential for numerous healthcare applications [109]. Several unique shapes of wearable body-worn sensing devices are proposed in the literature. In general, there are two types of sensing and monitoring systems: contact-based and contactless wearable sensor devices. They include optical and RF signal-based devices. Vital signals are frequently monitored by wearable sensors; nevertheless, there are drawbacks to this technology, including the high costs, dependence on user compliance, and the potential for disease transmission if not adequately disinfected. In situations where hygiene is a concern, like hospitals, contactless systems, especially those that use RF signals in the range of 30 kHz to 300 GHz, offer a viable alternative by enabling non-intrusive monitoring without direct touch or line-of-sight. Wi-Fi signals, which range in frequency from 2.4 GHz to 7.2 GHz, have become an effective means of tracking activity and identifying vital signs. These signals find use in fall detection and real-time health monitoring. By integrating these Wi-Fi-based devices with IoT frameworks, healthcare providers can receive useful data, and smart-home features like automatically altering environmental parameters depending on vital signs can be improved. But compared to wearable and visual-based systems, Wi-Fi and other contactless technologies have some limitations in terms of precision and dependability, notwithstanding their potential. 

It is anticipated that advancements in big data and machine learning, as well as the development of sophisticated semiconductor technologies like CMOS and Bi-CMOS, would lead to improvements in wireless sensing for medical applications. Technologies, including mHealth, 6LoWPAN, and Institute of Electrical and Electronics Engineers (IEEE 11073, Health informatics—Medical/health device communication), are being investigated for remote health monitoring in smart-city frameworks as IoT solutions in healthcare progress from basic designs to more elaborate smart systems [110]. Furthermore, new technologies like NFC, RF transmitters, and carbon nanotubes are opening new possibilities for data extrapolation in smart healthcare applications. With the help of multidisciplinary research, these technologies should continue to progress, opening new ways for contactless, affordable healthcare monitoring, especially when issues with accessibility and dependability are resolved [110,111,112]. Recently, wearable sensors for various purposes, including energy storage and health monitoring, have been transformed by recent developments in biodegradable and superhydrophobic materials-based and MXenes-based devices [24,86,113]. Furthermore, multimodal sensing e-skins fall into three categories, such as systems that combine chemical, electrophysiological, and physical sensors, and are investigated to simulate human skin for next-generation smart technology [5]. Recently, self-powered Triboelectric Nanogenerator (TENG) sensors monitoring ethanol during simulated exhalation and humidity using copper-oxide and polyethylene-oxide nanofibres have been investigated, which are suitable for non-invasive breath analysis applications. Key characteristics of the flexible and stretchable sensors created in this manner include their biocompatibility, scalability, lightweight design, flexibility, and wearability. Additionally, connection with a triboelectric source has made it possible to develop a self-powered, wearable, portable sensor. In conclusion, a wearable sensor for healthcare systems that runs on piezoelectric and triboelectric nanogenerators has been demonstrated. Smart sensors are useful because they can be created rapidly and affordably, worn conveniently, have a long lifespan, react swiftly, and may be used anywhere and at any time. They are also resistant to water and other factors [2,3]. 

However, SAR, which is regulated to prevent dangerous exposure, assesses the rate at which the human body absorbs energy from electromagnetic fields (EMF) in wearable devices. However, millimetre waves, which are now increasingly used in technologies like 5G, raise additional concerns [31]. Table 2 summarises various physiological signal-sensing devices.

## 3. Fabrication Methods

Fabrication methods play an important role in the accuracy and precision of the measurements and impose various limitations on the cost and fabrication time. Various fabrication methods have been utilised for wearable devices, which are summarised in [145]. There is a growing need for affordable and readily accessible fabrication methods that enable the mass production of wearable and flexible electronic devices [11,12,13]. DW is an AM method that directly transfers onto a substrate material and that can be used in various industries, including microelectronics, optics, and biomedical engineering. DW is a flexible fabrication method that eliminates the need for moulds or masks and allows for exact material deposition on substrates. Because DW makes high-precision patterning possible, it can be applied to the production of microdevices, PCBs, and biosensing. With the help of the extrusion based DW assembly technique, a large variety of inks may be patterned with feature sizes as small as 250 nm in both planar and three-dimensional forms (3D). However, compressed air is utilised in this procedure to force inks with regulated rheological characteristics via a single nozzle (diameter: 1 to 500 μm). The DW assembly uses a regulated printing speed and pressure that are dependent on the rheology of the ink and the diameter of the nozzle to deposit inks at room temperature or in an appropriate coagulation reservoir. DW assembly has, nevertheless, been used to handle a broad variety of inks, including colloidal suspensions and gels, nanoparticle-filled inks, polymer melts, fugitive organic inks, hydrogels, sol-gel, and polyelectrolyte inks. While some inks, like polyelectrolyte inks, must be written into reservoir-induced coagulation in order to facilitate 3D printing, other inks, like sol-gel inks, can be printed directly in air and offer superior control over the deposition process, allowing the ink flow to be repeatedly started and stopped during assembly by layer-by-layer extrusion of viscoelastic inks.

DIW, a subset of DW, enables the creation of 3D structures with unique shapes. Applications of DIW include energy storage devices, soft robotics, and tissue scaffolding. DIW, called direct-write assembly, is an extrusion-based 3D AM printing method in which liquid or semisolid colloidal inks are dispensed via small nozzles under controlled pressure and flow rates. Several DIW methods are available that can make 3D patterns inside the materials. They can be divided into the following two categories: droplet-based techniques like ink-jet printing and hot-melt printing, and filamentary-based techniques like micro-pen writing, robocasting (or robotic deposition), and fused deposition. 

Numerous ink patterns have been used, such as concentrated polyelectrolyte complexes, highly shear-thinning colloidal suspensions, colloidal gels, polymer melts, and diluted colloidal fluids. These inks solidify via solvent or temperature-induced phase changes or by liquid evaporation gelation. It is possible to create 3D structures with high aspect ratios (like parallel walls), spanning features, or continuous solids by carefully adjusting the ink composition, rheological behaviour, and printing conditions. Because the later structures require self-supporting elements to bridge gaps in the underlying layer or layers, they present the most difficulty for ink designers. To write 3D periodic architectures with filamentary features ranging in size from hundreds of micrometres to sub-micrometres, this feature article primarily focuses on our recent efforts to design concentrated colloidal, fugitive organic, and polyelectrolyte inks. These inks can be used as functional composites, microfluidic, and templates for photonic bandgap materials and inorganic–organic hybrid structures. Thanks to these recent developments, it is now possible to print metals and dielectrics simultaneously, overcoming earlier difficulties with print resolution and material compatibility. Examples of these uses include the construction of novel microelectronics, biocompatible medical scaffolds, and high-performance energy storage systems [27,28]. However, both DIW and DW are briefly explained, with some examples provided for clarity.

### 3.1. DIW Method

Writing plays a ubiquitous role in our daily lives, enabling us to document and communicate through various signs and symbols. As writing tools were invented and writing capabilities expanded, it significantly propelled human civilisation forward. In the context of designing electronic devices for wearable applications, Figure 4 shows four commonly utilised types of writing instruments: brush-type simple pen, pencil, fountain pen, and ballpoint pen [146]. Several writing materials, such as ink or graphite, are used to create signs and symbols on many substrates. Various pen-based techniques have been developed to fabricate electronics for wearable applications, including those utilising paper as a substrate. Each writing approach used in the creation of electronic devices has its own set of advantages and disadvantages. It is crucial to select an appropriate writing process based on the desired quality of electrical components and their specific application domains. Subsequently, the methods employed for device writing can be categorised into two groups: pen-based techniques, which comprise fountain pen, ballpoint pen, brush pen, and pencil methods; and pen-analogue approaches, as illustrated in Figure 5a–d. While the writing process for electronics using pen-based techniques is fast and portable, it is not the most convenient or cost-effective method due to the challenges associated with post-treatment procedures. In certain situations, extended durations, high temperatures, and vacuum settings are necessary. Moreover, despite the ability to write various components, pen-based direct writing methods are limited in their capability to fabricate complex components and devices, such as integrated circuits.

Ideal electronics for diverse applications should possess qualities such as affordability, user-friendliness, high performance, and rapid prototyping capabilities. Recently, there has been great interest in the development of flexible electronics due to its many potential uses, including wearable functional gadgets [147], medical monitoring systems [148], and flexible energy storage and conversion [149]. For instance, a polyester-based integrated epidermal electronic system resembling a tattoo has been created, holding promise to enable doctors to timely monitor patient vital signs straightforwardly and effectively [150]. This system can sense and record real-time muscle movement and heart activities and detect brainwaves without expensive, bulky medical instruments. To create electronic devices that offer enhanced performance at a reduced cost, considerable attention has been devoted to the development of various elements closely linked to flexible electronics. These include substrates, conductive materials, and production processes. [151,152]. Numerous methods, including coating, sputtering, and printing methods such as gravure, flexography, inkjet, and screen printing, have been utilised to produce flexible electronics. Printing techniques allow for the rapid development of parts and gadgets with precise three-dimensional (3D) architecture, which makes it easier to create usable 3D electronics. As an example, the development of strain and pressure sensors in a stretchable silicon-based polymer substrate has shown a feasible solution in recent 3D-printed devices, showcasing a huge potential to produce sensing electronic systems with arbitrary shapes used in different wearable applications. Nevertheless, these advanced techniques may not fully address the increasing need for low-cost, easily accessible, and efficient prototyping of flexible electronics. This is due to coating processing times, high energy consumption associated with sputtering, and the expensive nature of required printing equipment [153,154]. These research activities represent a ground-breaking revolution in the advancement of circuit systems, sensors, energy devices, communication elements, and more. It encompasses a wide range of areas, including materials innovation and transformative manufacturing approaches [152,154].

Recently, a novel approach has surfaced for creating electronic circuits using handwriting, where an array of writing instruments, including brush pens, pencils, fountain pens, and ballpoint pens, are employed to directly write on dielectric substrates and create electronic devices. This alternative method has gained attention as a promising way to integrate traditional writing tools with electronic functionalities. This technique of direct writing with a pen offers a convenient and rapid method for depositing electrically conductive materials on substrates, benefiting new biomedical and electrochemical sensing applications, electronic components, and energy storage devices. Pen-based writing enables end users to create and deploy sensor devices on-site as per their requirements. Additionally, when combined with printing methods, pen writing offers simplicity, cost-effectiveness, mass production capabilities, and high-resolution printing, known as the pen-analogue writing method. This method primarily encompasses applications like electrohydrodynamic and micro-plasma-based drop-on-demand (DoD) techniques. An example of a precise and versatile pen plotter, the AxiDraw machine, is demonstrated in Figure 6, which is used to design various items such as postcards, invitations, or drawings [155]. The researchers have used hand-writing electronics (HWE), which can provide the same benefits as printed electronics (PE) at a lower cost, making it simpler to prototype and create do-it-yourself (DIY) electronics. Figure 6 demonstrates mechanical printing processes, in contrast to conventional direct painting or writing methods, which enhance the accuracy of liquid metal-based flexible electronic circuits, a method that can be utilised for healthcare applications and wearable devices [156,157].

A variety of electronic devices use silver nanoparticle ink [158]. For example, to print on a mould surface, an automated micro-pen type dispensing device (such as Nordson Pro4 EFD) was utilised. After printing, the mould and printed traces were subjected to a 30-min sintering process in an ISOTEMP 282A vacuum oven [158]. The printed traces were positioned at a specific distance from the injection point on the mould surface to investigate the impact of polymer pressure on the interlocking and adhesion of ink/polymer. Each trace was printed in an inline form, measuring 20 mm in length and 50 μm in height. As discussed in detail in [158], a pen-based 3D printer provides a direct 3D structure, but using a pen in space and on a 2D surface is not convenient. Recently, researchers created a pen-based 4D printing (4Dp), which includes dipping the devices in a monomer solution to turn 2D prints into 3D shapes. One material used in 4Dp is MXene inks, which are produced from titanium carbide (Ti3C2) sediments that resemble clay. Ti3C2 inks can be utilised for stamping, printing, painting, and writing [159]. A pen-type setup was utilised to design glass nozzles using a micropipette puller (Sutter-based P-1000 instrument, as the company is located in Novato, CA, USA) that provides varied apertures, as shown in Figure 7 [160].

Direct laser writing (DLW) offers the capability to create precise patterns without the need for masks or chemicals. In a specific application, DLW was employed to fabricate a three-dimensional (3D) print of reduced graphene oxide (RGO). The researchers utilised a programmable approach that involved scanning a laser beam voxel-by-voxel, with each voxel measuring approximately 300 nm in diameter. By focusing the laser on concentrated flakes of graphene oxide (GO), the overlapping RGO flakes formed a 3D-printed structure. The treated area exhibited enhanced photoluminescence compared to the untreated region, enabling clear visualisation of the 3D pattern created by DLW [161]. Cellulose paper with microscale fibres can be used as a substrate for flexible-type organic light-emitting displays. Various facile pen-drawing can be used. Pens offer the versatility to utilise a variety of conductive and light-emitting materials for designing anode and hole-type injection layers. These materials can include different transfer layers, emission electron injection layers, and cathodes [162].

### 3.2. DW Methods

Three-dimensional printed designs utilise printing layer-by-layer and additive manufacturing methods (AMs). Three-dimensional printing (3Dp) has various advantages as compared to other methods. It provides design flexibility and high resolution as well as low cost [163]. AM is an advanced technique that encompasses various methods such as fused deposition modelling (FDM), selective laser melting (SLM), electron-beam melting (EBM), aerosol jetting, inkjet printing, and more, which have become increasingly accessible in recent times. The new generation of AM techniques allows for the utilisation of a wide range of materials, including polymers, metallic nano inks, ceramics, nanocomposites, and alloys. These AM methods enable the production of flexible sensor devices. Notably, extrusion methods like FDM and FFF (fused filament fabrication) are commonly employed in AM for creating 3D designs. Some examples are depicted in Figure 8, including a rigid device shown in Figure 8a [164], a flexible device shown in Figure 8b [165], and a microsensor shown in Figure 8c [166,167].

For the successful printing of complex and 3D surfaces, precise control of the laser beam is crucial. To achieve this, a shutter is typically placed in front of the nozzle to modulate the beam. Additionally, maintaining an optimal distance between the nozzle and the substrate is essential, typically ranging from 1 mm to ≥10 mm. Deviating from these specified boundary conditions can lead to undesirable overspray defects in the final design [168,169]. Three-dimensional printing technologies have significantly impacted the biomedical field, emerging as an innovative method with diverse applications such as tissue engineering, organ fabrication, medicine, drug delivery, and more. In the biomedical domain, various 3Dp methods are utilised, including FDM, selective laser sintering (SLA), and DIW. These methods make use of a wide range of materials, which are summarised in Figure 9 [164].

Three-dimensional printing is utilised in the fabrication of various medical devices and structures, such as patient-specific orthoses, prostheses, craniofacial implants, and medical devices based on individual data. Three-dimensional printing utilises low-melting-point metal ink in an additive mode, offering versatile applications across various fields. In [170], liquids such as gallium, bismuth, and indium alloy were used to fabricate 3D devices. In the realm of 3D devices, certain inks such as Bi35In48.6Sn16Zn0.4 exhibit unique characteristics wherein they do not absorb or release heat while behaving like metals during phase changes. Leveraging these novel ink properties, along with silicone rubber (a nonmetal), hybrid 3D-printed structures for wearable applications have been successfully fabricated. These structures possess remarkable mechanical and electrical properties while also exhibiting thermal conductivity. The exceptional insulating properties of the non-metal ink contribute to the durability and functionality of the 3D-printed designs, ensuring their effective performance even in demanding environments. Three-dimensional printing offers the capability to create a diverse range of antennas, encompassing polymer, metallic, ceramic, composite, and integrated antennas with multiple materials. For instance, an e-band lens antenna was designed using an automatic synthesis algorithm and realised through polyjet 3D printing technology using VeroWhite materials [171]. Furthermore, the field of electronics, sensors, and electrochemical devices has seen the utilisation of several conductive polymers and nanocomposites, enabling advancements in their respective applications [172].

Various 3D-printed multilayer-phased array antenna systems have been designed for WBAN applications [173]. In [174], a common freeze-casting technique was used to fabricate liquid-based 3D antennas. Researchers tested micro components of the liquid metal alloy for 3Dp (at 30 °C). Additionally, liquid metal has emerged as a viable option for creating injectable 3D bio-electrodes used in ECG and EEG measurements. Advanced, high-resolution, 3D-reconfigurable liquid-metal structures with stretchable properties have been designed to integrate seamlessly into the human body, as depicted in Figure 10a [175]. This design enables the creation of reconfigurable antennas that can be tuned by altering their structures, as well as reversible, movable interconnections that function as mechanical switches. In the realm of 3D printing, the intricate housing structure of the antenna, which may include components like a microfluidic channel, can be fabricated directly onto the substrate using primarily the FDM technique, as illustrated in Figure 10b. Furthermore, researchers have developed a heat sink composed of silicon micro pin fins to efficiently cool a Stratix 10 GX-based field-programmable gate array (FPGA) that consists of multiple heterogeneous dice [176]. To enhance the thermal performance of a design, a piece of 3D-printed plastic with an embedded silicon micro pin-fin heat sink was used to seal the tops of the pin fins and connect the heat sink to the inlet and outlet tubing.

Various 3D-printing methods of liquid metal are available to design antenna and electronic devices, such as liquid-metal melt deposition printing, liquid-phase printing, suspension printing, microcontact printing, and in vivo 3D-printed moulding methods [177]. An ultra-compact implanted antenna that is integrated with a Nanostim leadless cardiac pacemakers (LCP) system is presented for use in biotelemetry [178]. A compact antenna that can work in the 2.4 GHz ISM band is created for integration with an LCP. Wearable antennas that are primarily functioning in *ultra-high frequency* (UHF) and microwave bands have been developed using AM [179]. In [180], it is reported that 3D polymer-conductive hybrid materials were designed using the 3D-FDM method that can be used in wearable applications; an example is shown in Figure 10c. RFID antennas may be designed utilising a 3D-DW method on a textile material, for example, see the design reported in [181]. In recent developments, the integration of FDM for the substrate and syringe dispensing for metallic layers has enabled the creation of a 5G antenna embedded within a medallion [182]. It is suggested to employ removable fingernail-mounted 3D mm-wave and microwave antennas for on-body communications [183]. For this purpose, EGaIn was utilised as the conductive part of the radiator, while Visijet M3 crystal resin was employed as a dielectric for 3D-printed planar and helical antennas [184]. A groundbreaking approach involves the utilisation of 3D printing to create new and innovative gradient refractive index (GRIN) lenses for an H-plane horn antenna [185]. These lenses are constructed using high-permittivity ceramic ZrO_2_, resulting in enhanced performance and functionality. As reported in [186], a 3D-printed wire electrostatic discharge (WESD) structure comprising six upper capacitors was designed, enabling verification of the curved surface. This demonstrates the capability of 3D-printing technology to seamlessly integrate WESDs into various wearable devices, including sensors, energy harvesting systems, and light emitting diodes (LEDs), as shown in Figure 10d.

Utilising the DIW method and the PEDOT: PSS solution, researchers have successfully developed a high-resolution 3D-conducting polymer. This conducting polymer, along with its composites, has been extensively employed to construct flexible electrodes in diverse applications. The DIW method, in combination with the PEDOT: PSS solution, offers a versatile approach for fabricating flexible electrodes with exceptional precision. By precisely depositing the conducting polymer onto desired substrates, researchers have been able to create intricate electrode structures with enhanced conductivity and flexibility [187]. An innovative double-layered graphene conductor has been developed to enhance the optical and electrical properties of organic LEDs, offering a high surface area for improved performance [188]. A novel method called consecutive ink writing (CIW) has been developed to enable the printing of conducting polymer composites onto various flexible plastic substrates. This method involves an iterative extrusion process that produces a filament with a consistent diameter of 1.75 mm. By incorporating matrix polymers and carbon fillers, composite materials can be effectively used in 3D-printing applications. The CIW method provides a versatile approach for fabricating conducting polymer composites with precise control over the deposition process. The iterative extrusion ensures a uniform filament, which enhances the consistency and reliability of the printed structures. This method allows the integration of carbon fillers, enabling the customisation of electrical conductivity and mechanical properties of the printed composites.

**Figure 10 sensors-25-01377-f010:**
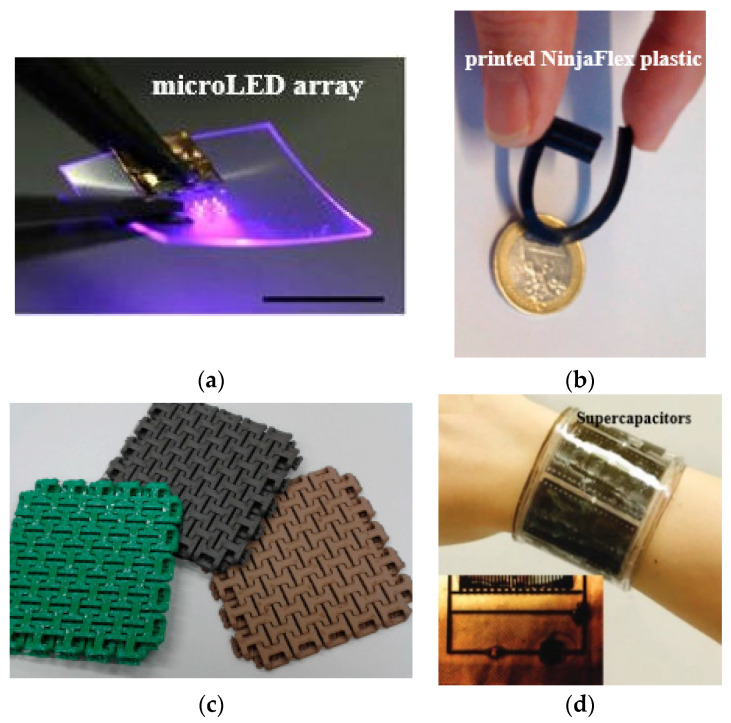
3D designs: (**a**) micro-LED array [175], (**b**) fluidic antenna [184], (**c**) hybrid materials [180], and (**d**) six capacitors [186].

The ability to print conducting polymer composites on flexible plastic substrates opens a wide range of possibilities in the field of flexible electronics. These composites can be used to create functional components, such as sensors, electrodes, and interconnects, with enhanced electrical conductivity and mechanical flexibility. The versatility of the CIW method enables the fabrication of complex 3D structures, further expanding the potential applications of these composite materials in various industries, including electronics, robotics, and healthcare [189]. This novel CIW process makes it possible to quickly print composite materials with a lot of graphene filler onto different substrates in 2D and 3D shapes. The CIW method allows for 3D printing of PEDOT: PSS and graphene ink on both rigid glass and flexible polyimide substrates. This versatile technique offers precise control over the deposition process, enabling accurate and reliable printing of conducting polymers and graphene materials. The CIW method has significant potential in various fields, such as flexible electronics, sensors, and energy devices. Further research and development in this area will optimise the CIW technique, opening possibilities for creating complex and functional 3D-printed structures on different substrates; an example is shown in Figure 11a [190]. Recently, polymer 3D printing has been utilised in devices for healthcare applications; for example, tissue scaffolds have been designed using polyjet and stereolithography printing that can be used to achieve hierarchical forms that can mimic the bone. A conductive polymer can be used to create implants that do not degrade in the human body. Polymer 3D-printing methods such as extrusion, resin-based printing, and powder-based printing provide flexibility in material choices and support a wide range of design shapes, responses, layouts, etc. [191]. To create a textile-based sensing device to be used in healthcare applications, several designers have incorporated 3Dp technology, including FDM and SLS; an example is shown in Figure 11b [192].

Spray coating is a DW method that can make a film deposition on a substrate, and which changes its dielectric values (ε) (Figure 12). This method has been broadly utilised because of its high speed (200 mm/s). Inks with a wide range of viscosities from 0.7 to 2500 mPa can be printed. Antennas, which can be used on IoT devices, have been easily fabricated using a simple spray-on method. As an example, researchers fabricated a 1-mm-thick MXene RFID tag with a reading range of 8 m at 860 MHz [193]. In [194], a liquid metal was sprayed to shape conductive paths on the elastomeric substrate. However, a textile duplex antenna utilising knit, and nonwoven fabrics was designed. The dielectric value $\left(\varepsilon \right)\;$of the fabric was enhanced due to a DW spray coating using nanoparticle ink.

### 3.3. Other Emerging Methods

Four-dimensional printing is a groundbreaking advancement in 3D-printing technologies that introduces the concept of self-transformation and functionality in printed materials. As illustrated in Figure 13a, 4D printing enables materials to change shape and perform specific functions in response to stimuli such as osmotic pressure, heat, current, ultraviolet light, and various energy sources. Unlike traditional 3D printing, which operates in three dimensions (*x*, *y*, and *z* axes), 4D printing adds a fourth dimension: time. This new dimension allows for dynamic and programmable behaviour in printed objects, revolutionising the design possibilities and applications of 3D-printing technology [195]. This fourth dimension allows for dynamic changes in the geometry of printed objects in response to external stimuli, thanks to the unique properties of smart materials. The design process of 4D printing enables the creation of objects that can undergo shape transformations and adapt to their environment, enhancing their functionality and versatility [196]. The development of 4D-printed devices involves the selection of various components, including printing technologies, materials, structures, stimuli, and applications. One of the key features of 4D designs is their ability to change shape, which can be achieved through three different approaches: using stimuli-responsive materials, employing a multi-material, or artificially inducing internal tensions [197].

The first approach, utilising stimuli-responsive materials, is widely popular and often considered the primary mechanism in 4D printing. In this technique, the 4D-printed structure initially possesses a temporary shape, and it transforms into its intended permanent shape when exposed to the appropriate stimulus. The second approach, known as multi-material, structures, leverages the unique characteristics of different materials, such as elasticity, swelling ratio, or thermal expansion coefficient, to achieve shape-changing capabilities. The multi-material structures combine many materials with varying properties, giving them special qualities, as discussed in 3D- to 6D-printing technologies. Different thermoplastics, polymers, and smart materials, including hydrogels and shape memory polymers (SMPs), can be combined with these structures. Combining these materials allows the production of objects with dynamic behaviours, such as the capacity to react to mechanical forces, temperature changes, and pressure, in addition to strength and durability. In biomedical applications, implants or devices must adjust to shifting biological conditions or mechanical stresses. The use of smart materials is the main factor driving the mechanism of form change in these multi-material structures. In 4D- and 6D-printing methods, these materials are designed to undergo controlled deformations in reaction to external stimuli. SMPs, for example, can be configured to hold a given shape and then revert to that form in response to a given stimulus, like heat or light. In contrast, hydrogels can expand or contract in response to variations in temperature or humidity. The capacity of these materials to recall and return to their original form when necessary is a crucial component of this shape-changing mechanism. This characteristic is essential for medical applications, like the creation of implants that may alter shape in response to changes in the surrounding tissue or the patient’s growth. The integration of smart material science and additive manufacturing techniques is the foundation of these technologies’ entire operation. The first step in the process is 3D printing, which constructs an object layer by layer. With the introduction of other dimensions like time and multi-directional production, this develops into 4D and 5D printing. The process of 6D printing combines the dynamic, stimulus-responsive properties of 4D printing with the structural benefits of 5D printing by employing five degrees of freedom to produce complex structures that can alter over time. Because of their intricate geometries, the printed items may now interact with their surroundings and have enhanced strength. This opens new possibilities in terms of self-healing, adaptation, and improved medical functions [139,140,141]. The third method involves intentionally inducing stress mismatching between layers, either during the design phase or during the printing process itself.

By carefully modifying the design and printing parameters, complex mechanical changes can be achieved when the stimulus is applied, effectively releasing the accumulated mechanical pressures [196]. Polyurethane was employed as a material for designing shape memory polymers that exhibit a time-dependent change, triggering cellular response and morphological transformations [198]. Another approach discussed in [199] involves the development of a thermally responsive theragripper using biocompatible and biodegradable poly materials. This theragripper is designed to deliver drugs through its layers and pores. Four-dimensional printing has been implemented using various DIW fabrication methods, making it accessible and compatible with a wide range of materials. The use of 4D printing has enabled the creation of smart, multi-material designs that are highly flexible, deformable, and efficient, with a self-enhancing capability. Numerous examples demonstrate the ability to achieve temporary and recovered morphologies in 4D-printed structures through diverse printing techniques, as illustrated in Figure 13b [200]. In [201,202,203], researchers developed more sophisticated structures through the combination of 4Dp materials. For example, these materials were made of nylon laminated with carbon fibre. They presented a broad range of frequency BW and radiation characteristics that can be utilised for 5G flexible and wearable devices. 4D printing is an innovation of a current 3D-printing method by incorporating time. However, 4D shapes can be deformed along with time upon the stimuli when they are exposed. Four-dimensional materials can respond to water, heat, magnetic and electric forces, light, and stress, as well as strain, pressure, etc.

**Figure 13 sensors-25-01377-f013:**
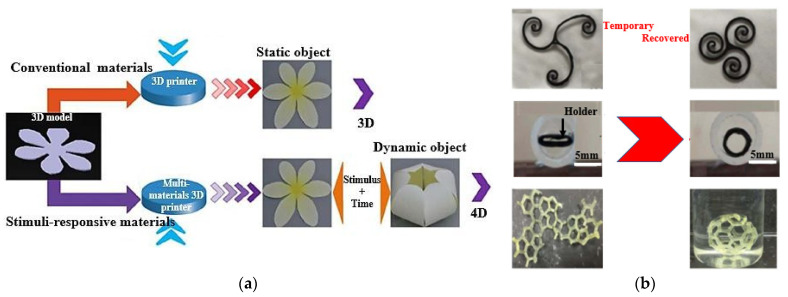
(**a**) 3Dp and 4Dp [197], and (**b**) temporary and recovered shapes of 4Dp shapes [200].

The research presented in [204] introduces a novel hybrid engineering process called 5D printing (5Dp), as depicted in Figure 14a, which combines additive and subtractive manufacturing methods. This technique enables the printing of intricate structures with three movement axes (dimension: x, y, and z) and two rotational axes. Five-dimensional printing has found applications in creating artificial bones for surgical procedures and food packaging. The printer utilises a printing head and two rotational printing platforms to accommodate the curved surface of bones and provide enhanced strength to the implantable bone structures. Comparative analysis shows that 5D printing is four times stronger than traditional 3D printing and can withstand greater pressure. This capability makes 5D printing suitable for complex surgical components used in surgery, surgical planning, teaching, and learning. Additionally, 5D printing has potential applications in the food-packaging industry for creating intricate boxes, tableware, and other complex designs. Figure 14b showcases the ability of 5D printing to design sophisticated and curved shapes. The versatility of 5D printing extends to automotive and medical applications, as highlighted in [205]. The equipment utilised in medical research needs to be stronger and have more intricate shapes. Five-dimensional printing can produce any curved surface, which can be employed in automotive and medical devices. For instance, a concave type of cap cannot be 3Dp; it requires a complex design, numerous fillers, and support. Five-dimensional printing facilitates printing by having the ability to print curved layers. Soon, 5Dp will surpass all 3D and 4D printing methods. Five-dimensional printing is a new method to design several orthodontic components, crowns, implanted devices, and accessories used in healthcare and dental surgery. Furthermore, 5Dp devices will be more durable than 3D- and 4D-based printing devices. The emerging technology of 5Dp holds significant promise for creating curved shape designs in various fields, including dentistry. Applications such as dentures, artificial bones, stents, and complex dental tools can greatly benefit from 5Dp’s ability to produce intricate and custom-shaped objects. Moreover, this innovative technology is expected to facilitate the development of wearable sensors for dental applications. With its unique capabilities, 5Dp is poised to enable the creation of complex wearable instruments that can enhance dental diagnostics and treatment. As a result, we can anticipate a wide range of successful applications for 5Dp in the field of dentistry and beyond.

A recent study conducted in Greece introduced the concept of 6D printing (6Dp) in the literature. Figure 15 showcases the integration of 4Dp and 5Dp technologies within the 6Dp process. The study suggests that 6Dp will utilise five dimensions to generate designs that possess intelligent properties and the ability to change shape in response to environmental stimuli. By leveraging the advancements made in 5Dp, the development of smart materials has paved the way for the creation of 6Dp products with unique and intelligent behaviour [206]. Moreover, the utilisation of 6Dp in the food industry holds promise for the creation of robust and highly responsive materials. In orthopaedics, 6Dp technology can be employed to develop smart casts that offer innovative solutions for treating conditions like clubfoot and bone fractures, including congenital talipes equinovarus. The utilisation of smart materials (SMs) in 6Dp technology allows for the precise application of mechanical loads, facilitating optimal alignment of fractured forearms or malleoli in orthopaedic applications. The smart cast can adapt its shape based on the presence or absence of oedema in the affected limb. This approach offers significant advantages, including the potential to eliminate the need for serial casting and the ability to customise the cast to gently guide the foot into the correct position for clubfoot treatment. Furthermore, the inherent flexibility of the 6Dp process, along with efficient setup configurations, may lead to reduced material consumption and faster processing times [204].

The concept for the new printing method, which combines the 4Dp and 5Dp, is depicted in Figure 15 [205,206,207]. To summarise, the emerging technologies of 4D, 5D, and 6D printing are currently being explored by researchers to enhance the design and production of intelligent structures using AM techniques. The integration of multi-dimensional capabilities with 4D printing offers the potential to create highly advanced smart objects, leading to significant advancements in medical and orthopaedic applications. The continuous investigation and improvement of these techniques will contribute to the overall impact of AM methods in various domains, benefiting society. Six-dimensional printing is a relatively new area of AM; it is anticipated to expedite the research into finding complex materials. It produces lighter, stronger designs that are more sensitive to their environment. However, the setup and operation costs for 6Dp are anticipated to be significant. DIW techniques will continue to advance toward diversifying the materials, architectures, and fabrication techniques. However, 6D printing offers a promising solution in various fields, such as healthcare, the food industry, and so on. In general, these fields are not fully explored and have a high potential for research as well as commercial applications. Combining conventional fabrication with newly developed DW and DIW in 4Dp, 5Dp, and 6Dp would be beneficial, as shown in Figure 16. Table 3 summarises the printed devices in the literature.

**Table 3 sensors-25-01377-t003:** Summarised 3D to 6D techniques and their characteristics are available in the literature.

*AM Methods/Types* *(References)*	*Equipment*	*Summarised Review* *(1) Equipment* *(2) Features* *(3) Limitations*
2010–2021 3Dp FDM, CJP, SLA, SLS, DLP, MJM [208,209,210]	3D printer	(1) Design layer-by-layer along a vertical axis (2) Provide tautness on the layer of the product and no intelligence
2013–2021 4Dp [211,212,213,214] 3DP methods+ SMP, SMA, LCE, Hydrogels	3D printer, intelligence materials	(1) Design smart products (2) Limits of layer intelligence, restricted programmable and directivity
2016–2021 5Dp [215,216] 3D CAD file, 3D Scanner, and 3D printing material	5D printers or robotic arms	(1) Design in every direction Less material and process time, (2) No intelligence, higher cost
2021~ 6Dp [215,217,218] 5Dp with smart materials	5D printers, intelligence raw material	(1) Layer in all directions, with less material and process time to design products., and easily programmable. (2) Higher setup cost and additional calibration required

**Figure 16 sensors-25-01377-f016:**
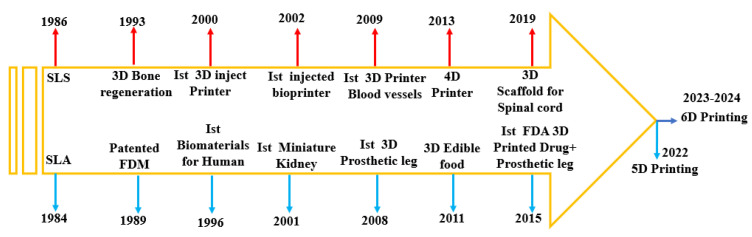
Previously designed 3D to 6D wearable and flexible devices in the literature review [219].

## 4. Human Body Models

In a body-centric device, it is necessary to assess the interactions between the human body and the device. In the initial research studies, the system was routinely evaluated with the help of tissue-equivalent phantom models. These phantoms could be numerical ones used in the simulation or physical ones used in experimental studies. These phantoms provided invaluable insight before moving to the animal or clinical study phases. This section provides insight into the state-of-the-art human body models utilised for the prediction of the body effects on the performance of wearable devices. Some examples in each area are provided for more clarification.

### 4.1. Type of the Human Body Models 

The link between the sensor antenna and the human body is critical for wearable devices. The heterogeneous body tissues are lossy, significantly impacting the sensor antenna and its performance. The anatomical differences between patients increase the complexity of a suitable design. Many factors should be considered for designing wearable sensor antennas; for example, their performance in the presence of a human body, and their interaction with it. Many human numerical and physical phantom models exist. The first safety concern is that the level of heat created due to radiated power by the antenna may be absorbed and converted to heat, increasing the tissue temperature. Therefore, the level of the input power should be precisely controlled. Commonly, the specific absorption rate (SAR) is calculated to determine the input power level that is safe [220,221]. Several rules and regulations for heat generated by radiated waves have been implemented based on scientific evidence to prevent any adverse effects on the human body. The American National Standards Institute (ANSI), the Institute of Electrical and Electronics Engineers (IEEE), and the International Commission on Non-Ionizing Radiation Protection (ICNIRP) established standards for the maximum permissible SAR. Table 4 summarises the references which calculated the SAR values based on these standards. Standard limits for the SAR values concerning frequencies are summarised in Table 5 and are calculated as follows by [222]:(1)$$SAR=\frac{{\left|E\right|}^{2}\sigma_{e}}{2\rho }$$
where E is the electric field strength (V/m), $\sigma_{e}$ is the effective electrical conductivity (S/m), and ρ is the mass density of the tissue (kg/m^3^).

#### 4.1.1. Simulation Based Human Body Models 

Near-field provides a strong coupling effect with the surrounding lossy media; as a result, the presence of a human body may affect the performance of a wearable device. An appropriate selection of the human body model is essential to accurately predict the interaction between the wearable device and the body. In the literature, various human body models have been shown and measured. In the numerical simulations, one may use a simple layered model of the body [223], like the one shown in Figure 17. These types of models, although they do not have the anatomical details of the body, are an effective tool for fast simulations and often are accurate enough for small devices. The advantage of a layered model is the fast simulation time that makes it possible to run optimisations for the best performance [224]. It is expected that after the simple simulations, a more realistic body model or an experimental study will follow to ensure the accuracy of the results. A good review of body phantoms used for the simulations and experimental validations in WBAN systems is presented in [225]. Multiple dielectric values of the biological tissues of the human body are briefly explained in [226].

**Figure 17 sensors-25-01377-f017:**
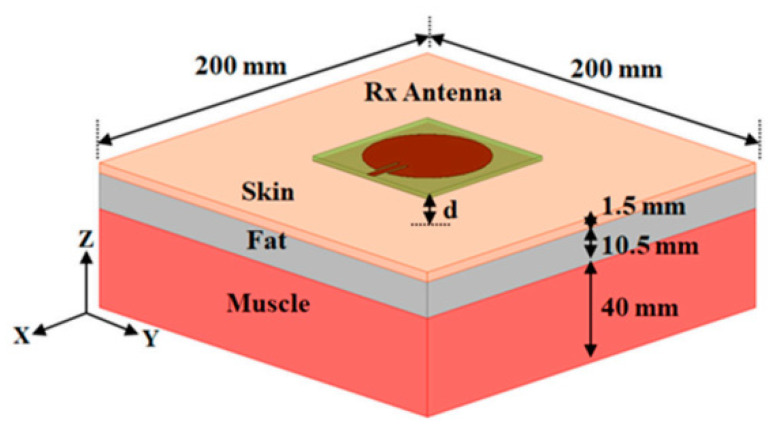
Layered model of the human body tissues used in the literature [227].

**Table 4 sensors-25-01377-t004:** A summary of SAR values calculations in the literature.

References	Models	Body Parts	Frequency (MHz)
[228,229]	Adult, adult	Head	900
[230]	Pregnant woman/fetus in the uterus	Torso/whole body	900, 2100, 2450, 2600
[231]	Adult	Whole body	98 to 2450
[232]	Adult	Head/torso	850, 900, 2100, 2600, 5100
[233]	Adult phantom	Brain/eye/skin	2300 to 2400
[234]	Adult/SAM	Head/brain/muscle	900, 2500, 3500
[235]	Adult	Brain/skin/fat/bone	900
[236]	Adult	Head/skin/bone	900, 1800
[237]	Adult, child	Whole body/head/torso	20 to 2400
[238]	Adult (male), child (Boy, girl)	Head/brain/eye	900, 1800
[239]	Adult (male, female), child	Head/brain	900, 1800, 2100, 2400
[240]	Adult, child	Head/fat/skin/bone/brain/eye	900, 800, 1800, 2100

**Table 5 sensors-25-01377-t005:** SAR limits for healthcare applications (W/Kg) [217,219].

Standard	Cube Mass (W/kg)	Whole-Body Average	Head and Trunk	Limb
**ICNIRP**	10 g	0.08	2	4
**IEEE**	1 g	2	1.6	4

Many factors, including application, computation time, and resource constraints, influence the choice of body models. It is usual to choose a theoretical or canonical phantom in the early stages of a design system for the following two reasons: to save time and computational resources. A more complex assessment phase will require realistic and voxel phantoms. Therefore, the simulation tools that use an accurate body phantom should not be underestimated. Another fact to consider is that fluctuations in the electrical properties of the body tissues may lead to changes in performance. This dependency on the body model and the type of phantom employed calls for a design that provides good isolation from the body and is not significantly affected by the body’s dielectric properties [241]. Some phantom models and existing animal tissues employed in the literature are depicted in Table 6. In the literature, simple geometrical shapes are utilised to evaluate the antenna performance in human body models. For example, rectangular cuboids are commonly used phantom models because they are simple and planar [242]. Cylindrical [243] and spherical phantoms [244] are often used to model the arm [245], neck [246], head [247], or eyeball [248]. In [249], ellipsoid shapes are used to represent the trunk. In [250], semi-spherical phantoms represented the breast models and were used to model hyperthermia treatment devices. In [251], canonical forms were used to model the entire human body. In [252], different geometries were used to model different parts of a pregnant female body. Various important factors in the design of wearable sensor antennas include the shape of the phantom models, the distance between the phantom surface and the sensor, the orientation of the sensor to the body surface, the frequency bandwidth, temperature, and tissue.

#### 4.1.2. Voxel Phantoms

Voxel phantoms represent more realistic shapes and arrangements of biological tissues. Various medical imaging systems have been used to provide such realistic human-body phantoms. One of these models is named “Hugo”, which was based on the visible human project in 1994, and it was the first accessible voxel human model, as seen in Figure 18a [253]. Later, several human body models were created that included more high-resolution representations of the entire body. The Japanese male and female models [254] and the virtual family models [255] are illustrated in Figure 18b. Due to the complexity of the model that requires computational resources, whenever possible, it is common practice to restrict the body model to cut a specific section of the voxel model. This is especially possible when the device is small and is utilised outside of the body, such as mobile phones, microwave hyperthermia devices, or microwave imaging setups, as well as small wearable sensors [256]. These phantom models offer tissue data about the desired parts needed for applications. For the SAR computation of mobile phones, specific anthropomorphic mannequin (SAM) phantoms are used [257]. A volume shaped like a human head filled with material resembling the head tissues’ average dielectric characteristics is used to model SAM phantoms. Realistic voxel breast models were employed in [258] for modelling microwave hyperthermia breast cancer treatment, as seen in Figure 18c. Various types of breasts, based on the American College of Radiology categorisation, were considered. The breast models may also be utilised for microwave breast cancer imaging [253]. Recently, there has been some interest in improving personalised treatment by creating patient-specific body phantom models. Furthermore, commercial electromagnetic solvers offer integrated modules that reconstruct 3D human models from magnetic resonance imaging (MRI) or computed tomography (CT) scan data. For instance, Sim4Life-based supports the integration of the medical image segmentation (MIS) tool Set (iSEG), as well as the ALBA 3D Seg-based module, as shown in Figure 18d [11,12,13,14,30,259,260,261], which was developed to integrate into the CST microwave studio software to design button antenna system [262]. The reconstruction of the 3D human models enhances the patient-specific simulations for real wearable applications. However, these computational human phantoms (CHPs) and their uses have achieved tremendous advancements in recent decades [262]. For example, CHPs characteristics enable the analysis of variations in the performance of wearable antennas and sensors due to the body’s movement and help fabricate the wearable structures accurately [263,264,265].

#### 4.1.3. Experimental Phantoms

To validate the performance of fabricated wearable devices, experimental phantoms have been employed. These phantoms can be constructed using simple materials or derived from ex vivo or in vitro tissue. During measurements, the electrical and thermal properties of body models are influenced by the materials utilised, underscoring the significance of understanding these material properties. Various phantom shapes, including the human body head, abdomen, torso, and other anatomical regions, are utilised in measurements to replicate real-world scenarios [265].

#### 4.1.4. Man-Made Phantoms

Man-made simple phantoms are created using chemical materials and may be liquid-based or gel-based. A simple liquid-based phantom may be created using saline water and salt [267,268]. Sometimes, a cavity is filled with a homogeneous lossy medium. This is a phantom used for various standard testing that may be utilised for wearable or implanted sensing devices, or a polyvinyl alcohol (PVA) type phantom was utilized, illustrated in Figure 19a [269]. Various recipes can be found in the literature. The composition of gel-based tissue phantoms can be tailored to replicate specific human tissues, such as muscle, adipose tissue, or breast tissue [270]. By adjusting the concentration of gel components and incorporating additives, the electrical conductivity and permittivity of the phantoms can be controlled to closely resemble those of real human tissues. In [271], a method to create various breast tissue-mimicking phantoms is presented. Schmid and Partner Engineering AG (SPEAG) designed a body phantom and a head phantom model for electromagnetic evaluations (POPEYE), which covers a frequency range of 300 MHz to 6 GHz, mimic a real situation, as shown in Figure 19b [272], and Figure 19c [273]. Creating a real body phantom with anatomical details is a complex process due to its complexity, which includes heterogeneous media with dispersive materials that exhibit changes in the dielectric values over a wide range of frequency bands.

#### 4.1.5. Ex-Vivo Animal Phantoms

The ex vivo phantoms often utilise parts of animal bodies or minced meat. Several examples of the applications of such models can be found; for example, they have been used in validating microwave hyperthermia [274], microwave ablation [275], and implanted biotelemetry [276] devices. An example is shown in Figure 20, stress monitoring system (WSMS) that use multi-sensors system. Some studies have used biopsy tissue samples from human patients. One consideration is that after the tissue is excised from the body, the dielectric properties may change [274,275,276]. This variation should be considered in the assessment of the performance of the device. Preclinical testing for developing innovative treatments for diseases involves a combination of in vitro cell examinations and in vivo animal studies. However, it is important to note that outcomes observed in vitro often do not accurately reflect those observed in vivo. This might be brought on by the absence of the complex natural bone environment in vitro. A paradigm where cells are kept in their natural three-dimensional habitat is offered by ex vivo bone explant cultures.

**Table 6 sensors-25-01377-t006:** Different types of human body models.

Phantom Models References	This Table Summarises the Man-Made Models Used in the Literature
**Realistic model** **[264]**	Visible human project, Japanese male and female, virtual family, CST Microwave Studio voxel family, SAM phantoms, realistic breast phantom
**Liquid phantom models [55,76,277]**	Medium of the canonical shield, liquid, and gel, multilayers
**Animal tissue [278,279]**	Human body models, Minced meat, minced pork, piece of pork, porcine abdominal tissues, porcine eye, rat skin, chicken breast

## 5. Conclusions

In this manuscript, a comprehensive review of emerging technologies for designing wearable and flexible sensor devices utilised in WBAN is presented. We reviewed up-to-date design systems in WBAN systems, which can enable the next generation of wearable and flexible sensor devices. Advanced wearable sensors for many applications in healthcare, self-energy, and smart technologies have been made possible by recent advancements in biodegradable, flexible, and stretchable materials, including MXene ink and multimodal e-skin sensor devices. In the next step, various DW and DIW fabrication methods were reviewed, with examples provided. These fabrication methods included 3D- to 6D-based. These methods utilise a wide range of substrates and conductive materials to design complex structures across diverse domains. Next, to evaluate the effectiveness and performance of wearable and flexible sensor devices, an accurate body phantom model is essential at an earlier stage of the designing process. However, various voxel body and corresponding tissue models were discussed and presented, with examples provided for clarity. By incorporating these requirements, this manuscript offers a unique view that simultaneously considers three dynamic developments that have not been extensively studied together. Currently, wearable, and flexible sensor devices face key challenges related to design size, power supply, durability, and biocompatibility, as well as needed improvement in manufacturing processes for fully flexible and stretchable sensor devices. SAR, which is regulated to prevent dangerous exposure, assesses the rate at which the human body absorbs energy from electromagnetic fields (EMF) in wearable devices. Millimetre waves, which are now increasingly used in technologies like 5G, raise additional concerns. However, there is the possibility that the next-generation healthcare sensor devices will be small enough that they will connect with the terahertz (THz) frequency and will be used for sensing applications and cancer detection, etc. However, the researchers have shown serious concerned about the adverse health effects to use radiofrequency (RF) fields above 6 GHz, especially for the 5 G mobile phone network [280,281].

## Figures and Tables

**Figure 1 sensors-25-01377-f001:**
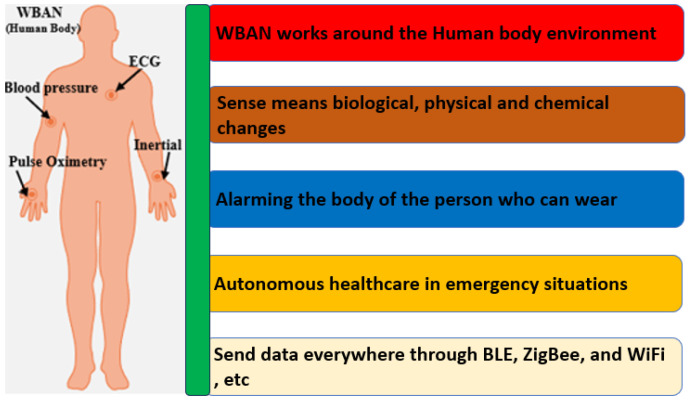
Examples of technologies used in WBAN systems.

**Figure 4 sensors-25-01377-f004:**
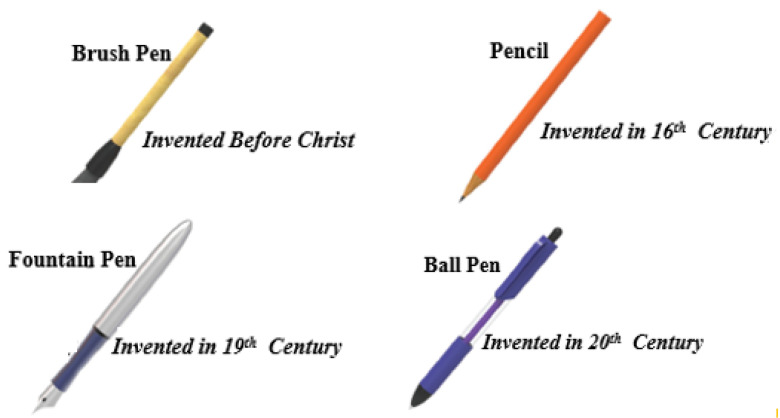
Development of direct-writing electronics for wearable and flexible applications [146].

**Figure 5 sensors-25-01377-f005:**
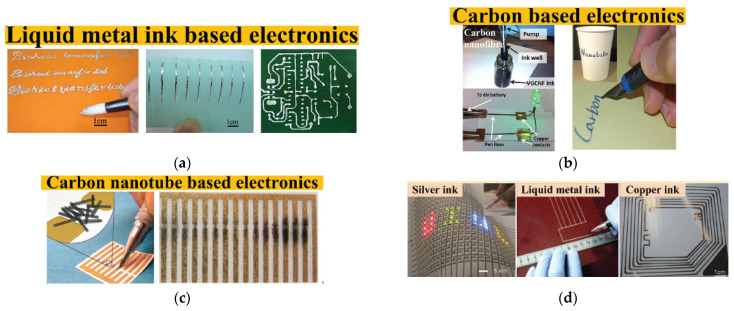
Writing electronics used in the literature: (**a**) brush pen, (**b**) fountain pen, (**c**) pencil, and (**d**) ball pen [146].

**Figure 6 sensors-25-01377-f006:**
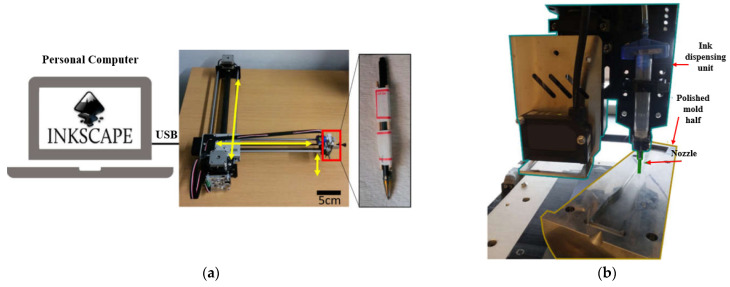
(**a**) Ultra-High Frequency (UHF) RFID tags [155], and (**b**) HWE system [157].

**Figure 7 sensors-25-01377-f007:**
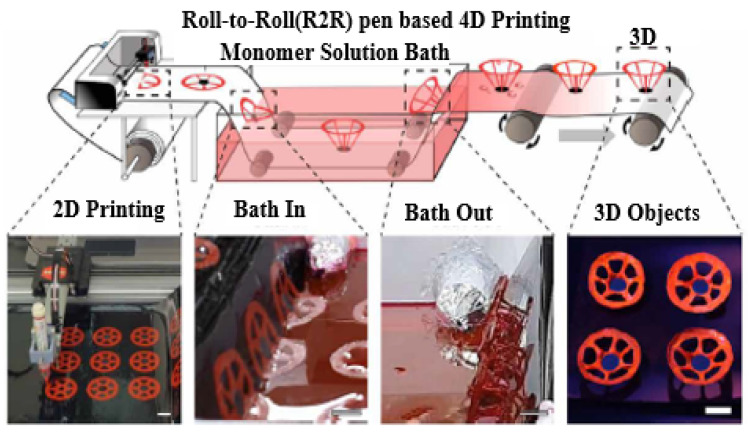
Flexible electronic: pen-based 4D printing [158].

**Figure 8 sensors-25-01377-f008:**
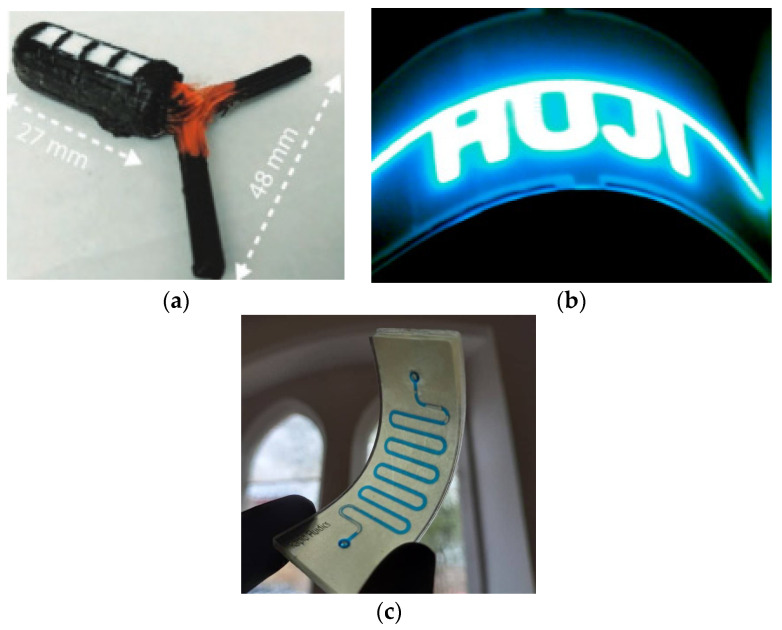
Flexible devices: (**a**) gradient echo device [164], (**b**) electroluminescence device [165], and (**c**) serpentine channel with integrated coverslips [167].

**Figure 9 sensors-25-01377-f009:**
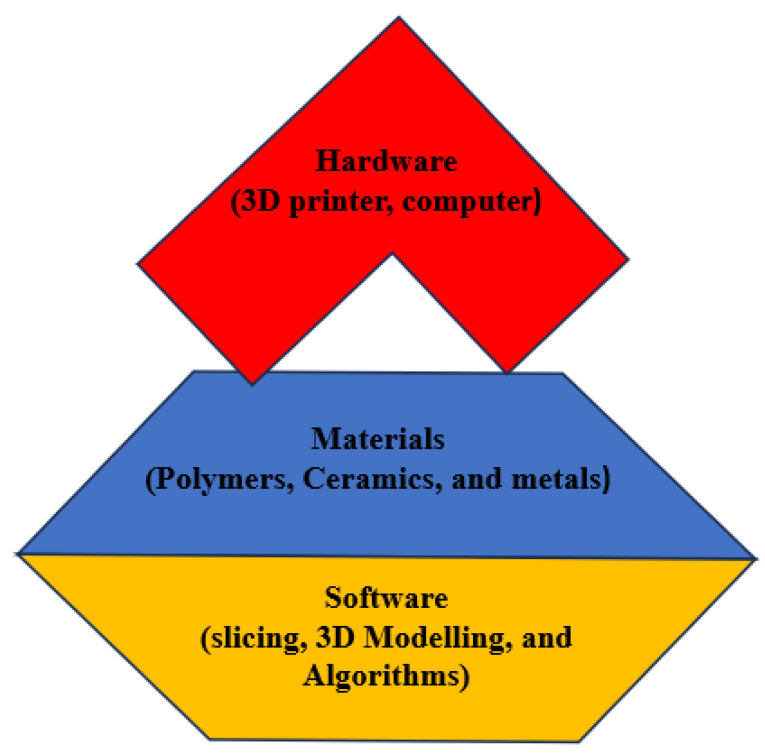
Fabricated devices using traditional 3D-AM methods.

**Figure 11 sensors-25-01377-f011:**
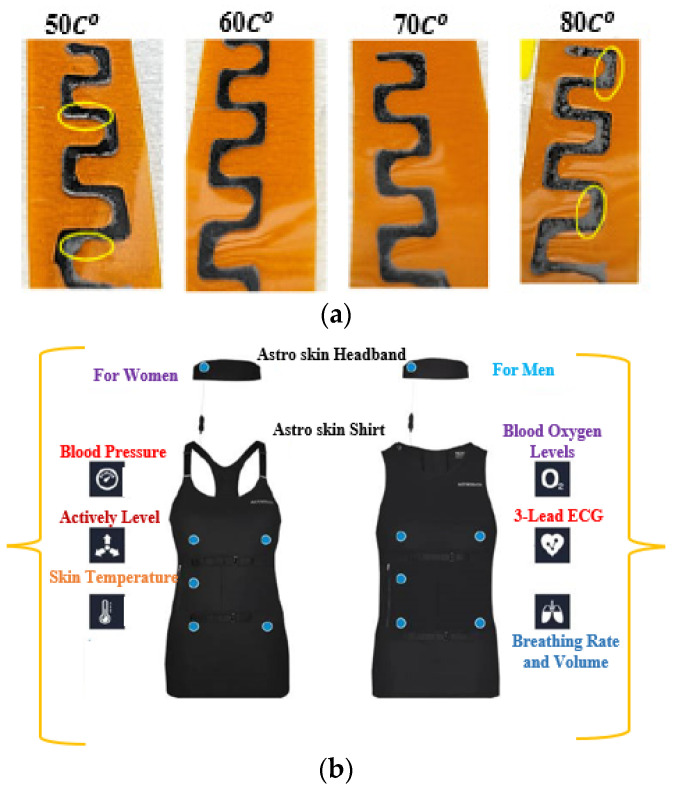
(**a**) EEG/PEDOT: PSS composite electrodes [190], and (**b**) flexible biosensor [192].

**Figure 12 sensors-25-01377-f012:**
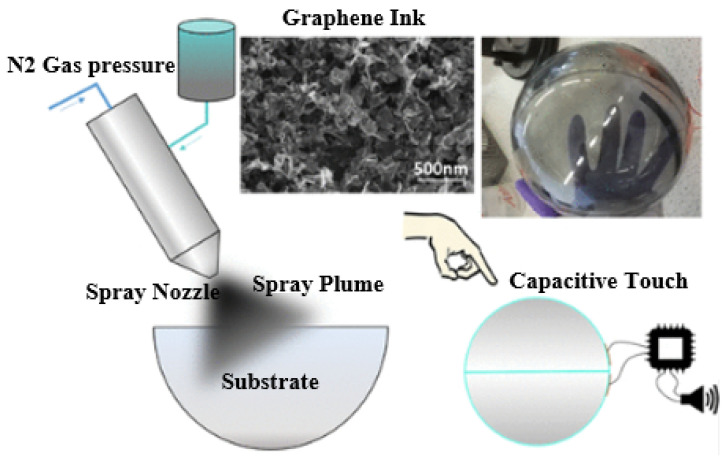
Spray coating process and semi-transparent capacitive-touch device [193].

**Figure 14 sensors-25-01377-f014:**
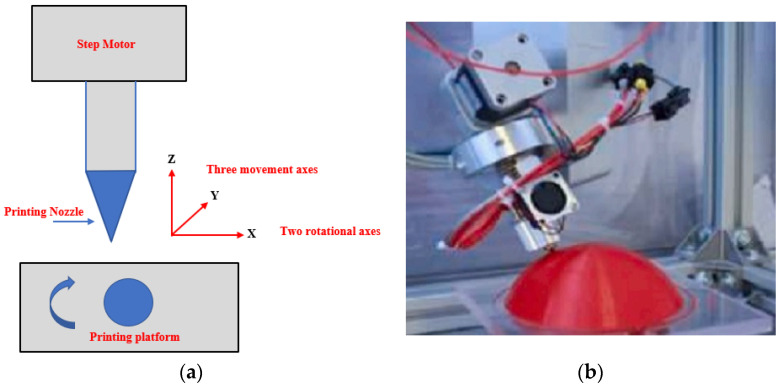
(**a**) Schematic diagram of 5D printing [204], and (**b**) printing of cap using 5Dp [204].

**Figure 15 sensors-25-01377-f015:**
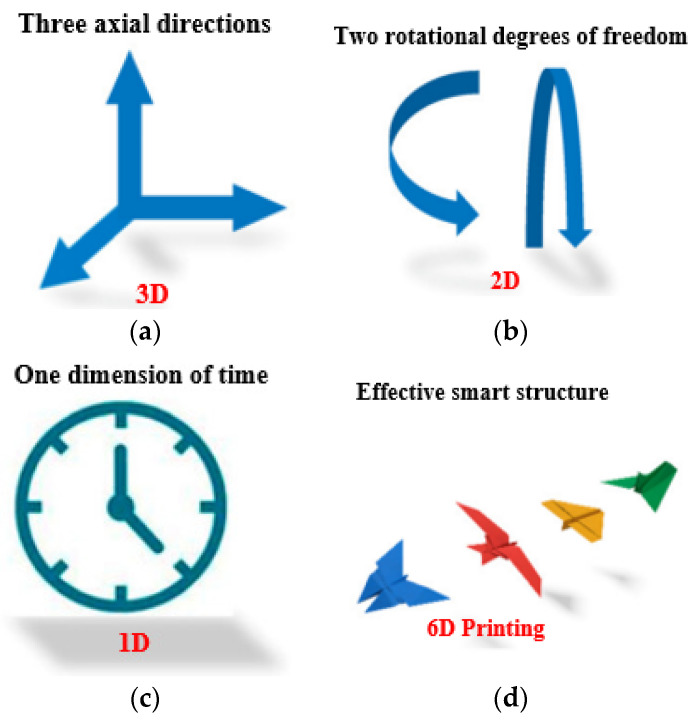
1D to 6D printing method in the visual (**a**–**d**): dimensionality clarification [205,206].

**Figure 18 sensors-25-01377-f018:**
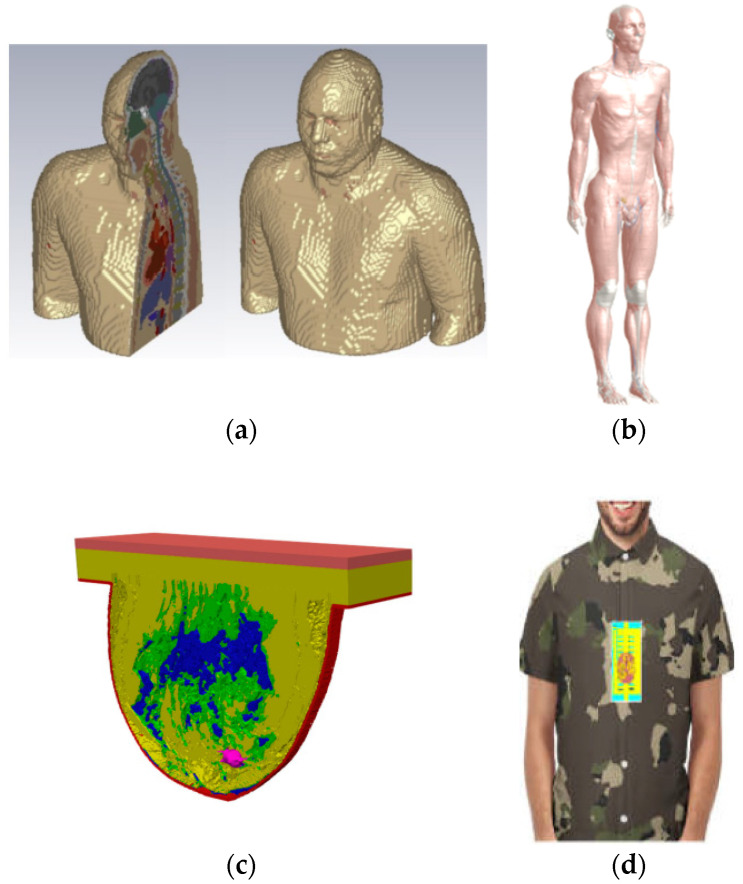
Human phantoms: (**a**) the Hugo [253], (**b**) the duke (virtual family) [255], (**c**) the breast model [266], and (**d**) button antenna test on 3D human body model [260].

**Figure 19 sensors-25-01377-f019:**
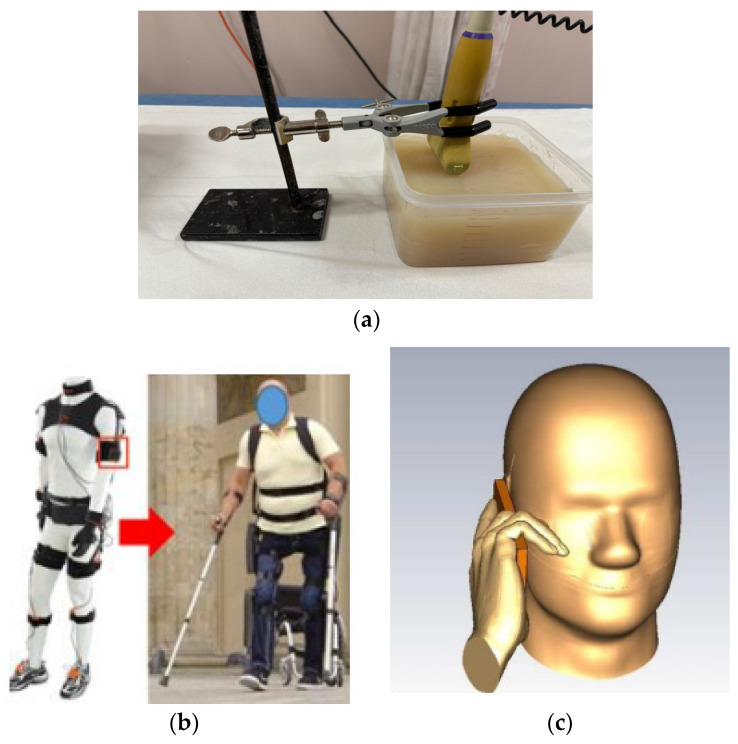
Human models: (**a**) multi-layers phantom [269], (**b**) device under test using human body with movable parts [272], and (**c**) SAM head [273].

**Figure 20 sensors-25-01377-f020:**
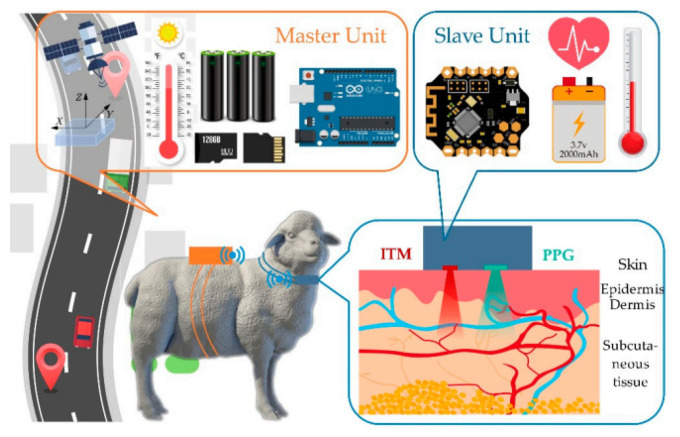
Examples of ex-vivo testing model [274].

**Table 1 sensors-25-01377-t001:** Overview of available wireless wearable technologies.

Tech/References	Frequency (GHz)	Range (m)	Power (mW)	Data Rate (kbps)
**RFID [35,36]**	0.1356, 0.86–0.96	0–3	200	640
**Bluetooth [37,38,39,40]**	2.4–2.5	1–100	2.5–100	1–3
**Bluetooth low energy [41,42]**	2.4–2.5	1–100	10	1
**ZigBee [43,44,45]**	2.4–2.5	10–100	35	250
**Wi-Fi [46,47]**	2.4–2.5	150–200	1 W	54
**UWB [48,49]**	3.1–10.6	3–10	250	53–480
**Adaptive network topology (ANT) [50]**	2.4–2.5	30	0.01–1	20–60
**Medical implant communication system (MICS) [51]**	0.402–0.405	2	25	200–800
**Infrared data association (IrDA) [52]**	0.00038	0.1	4–10	1 × 10^6^
**Near field communication (NFC) [53]**	0.1356	0.05	0.15 mW	424

**Table 2 sensors-25-01377-t002:** Various available wearable devices in literature.

References	Various Research Areas That Used Wearable Devices in the Literature
Healthcare/medical technologies for physiological signals monitoring systems	(1) Technologies (RF and microwave), (2) materials, and (3) applications and research directions
2017 to 2022 [113,114]	Wearable design and systems: healthcare, clinic, and hospital applications
2018 to 2022 [115,116]	Wearable design and systems: Energy harvesting, holistic systems, and self-healing composite
2021 to 2022 [117,118,119]	Next enabled device systems: Backscattering
2013 to 2022 [120,121,122,123,124,125,126,127,128]	Enabled device systems: RFID and RF sensing devices, far-field, and RF rectenna systems
2022 to 2023 [129,130,131]	Soft materials: Enabled electronic for medicine and human interfaces, and soft, skin-integrated multifunctional microfluidic systems
2013 to 2023 [132,133,134,135,136,137]	Liquid metal-based electronic: Sensing and multi-site sensing capabilities
2022 to 2023 [138,139,140,141,142,143,144]	Deep learning-based: Blockchain and artificial intelligence technology

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
