# Peer review of "Wearable and Flexible Sensor Devices: Recent Advances in Designs, Fabrication Methods, and Applications"

_sensors, 2025, doi:10.3390/s25051377_

Round 1

Reviewer 1 Report

Comments and Suggestions for Authors

This manuscript of review considers recent advancements in wearable sensor devices used in Wireless Body Area Networks (WBAN) and discusses the development of Direct Ink Writing DIW and DW technologies that have led to the creation of new, high-resolution, flexible, and stretchable wearable sensors.

1. In Page 4, there's a "DIW)". In Page 10, there's a repetitive citation of [128]. Please check the overall paper in detail in case any writing mistakes.

2. There's no specific DW and DIW definition before using it as the context, especially in Intro. Please add more background to make reader clear about what your paper emphasis.

3. Please compare with other review in this area, to show what's your specific view which differs yours from others.

4. Regarding to the wearable sensors for healthcare, please add the comparison review paragraph with remote sensing to enhance the convincing. Please consider the below papers for comparison refences: 

[1] Ge, Yao, et al. "Contactless WiFi sensing and monitoring for future healthcare-emerging trends, challenges, and opportunities." IEEE Reviews in Biomedical Engineering 16 (2022): 171-191.

[2] Li, Changzhi, et al. "Overview of recent development on wireless sensing circuits and systems for healthcare and biomedical applications." IEEE Journal on Emerging and Selected Topics in Circuits and Systems 8.2 (2018): 165-177.

[3] Alshamrani, Mazin. "IoT and artificial intelligence implementations for remote healthcare monitoring systems: A survey." Journal of King Saud University-Computer and Information Sciences 34.8 (2022): 4687-4701.

Comments on the Quality of English Language

Many writing mistakes due to careless. Author should pay attention in proof-reading.

Author Response

Dear Reviewer 1, thank you so much for carefully read our paper and gave us important comments. We have briefly addressed your comments. Once again, thank you so much.

Reviewer 2 Report

Comments and Suggestions for Authors

This manuscript provides a comprehensive review of wearable smart devices and their applications in recent years and is modeled using different human body models. A comprehensive review of various fabrication methods for wearable sensors for WBANs is also presented, discussing new challenges and future research directions. In addition, DIW and DW methods are introduced to provide new ideas for the preparation of high-resolution integrated smart structures. Thus, it is recommended that this manuscript be received with minor revisions to the following questions.

1. In the manuscript, lines 31-33 of page one describe the types of items in which sensors can be integrated. However, the author lacks access to recent articles in this area. It is recommended that the introduction be updated to include recent reports of relevant literature (https://doi.org/10.1016/j.cej.2024.154443 ) to provide a more comprehensive overview.

2. On page 2, lines 96-99, it says that the processing method of neural networks is used to improve signal accuracy. It is recommended that the authors read more research on this subject to make the overview more complete.

3. Page 3, lines 113-116 in the manuscript talk about safety issues regarding microwave exposure. It is recommended to read the relevant article and discuss this area in more depth.

4. Lines 710-713, page 18 of the manuscript, describe the properties of multimaterial structures. However, the mechanism of shape change is not described in sufficient detail. It is recommended to read the related article for a more in-depth explanation of the mechanism.

5. Some of the images in the manuscript are too low resolution (Figure 2, Figure 5, Figure 8c and Figure 18), which makes it difficult to read the content of the images. It is recommended that high-resolution images be used.

Author Response

Dear Reviewer 2, thank you so much for carefully read our paper and gave us important comments. We have briefly addressed your comments. Once again, thank you so much.

Reviewer 3 Report

Comments and Suggestions for Authors

Author Response

Dear Reviewer 3, thank you so much for carefully read our paper and gave us important comments. We have briefly addressed your comments. Once again, thank you so much.

Round 2

Reviewer 1 Report

Comments and Suggestions for Authors

The revision looks fine and answers all questions. The manuscript is well for publication.